# Spatially explicit capture recapture density estimates: Robustness, accuracy and precision in a long-term study of jaguars (*Panthera onca*)

**Bart J. Harmsen** [1,2]*, **Rebecca J. Foster**[1], **Howard Quigley**[1]

**1** Panthera, New York, New York, United States of America, **2** Environmental Research Institute, University of Belize, Belmopan, Belize

* bharmsen@panthera.org

**Data Availability Statement:** Data available in Supporting Information csv files.

**Funding:** BJH, RJF, and HQ received funding from Panthera to carry out this work from yearly budget.

## Abstract

Camera trapping is the standard field method of monitoring cryptic, low-density mammal populations. Typically, researchers run camera surveys for 60 to 90 days and estimate density using closed population spatially explicit capture-recapture (SCR) models. The SCR models estimate density, capture probability (g0), and a scale parameter (σ) that reflects ranging behaviour. We used a year of camera data from 20 camera stations to estimate the density of male jaguars (*Panthera onca*) in the Cockscomb Basin Wildlife Sanctuary in Belize, using closed population SCR models. We subsampled the dataset into 276 90-day sessions and 186 180-day sessions. Estimated density fluctuated from 0.51 to 5.30 male jaguars / 100 km$^2$ between the 90-day sessions, with comparatively robust and precise estimates for the 180-day sessions (0.73 to 3.75 male jaguars / 100 km$^2$). We explain the variation in density estimates from the 90-day sessions in terms of temporal variation in social behaviour, specifically male competition and mating events during the three-month wet season. Density estimates from the 90-day sessions varied with σ, but not with the number of individuals detected, suggesting that variation in density was almost fully attributable to changes in estimated ranging behaviour. We found that the models overestimated σ when compared to the mean ranging distance derived from GPS tracking data from two collared individuals in the camera grid. Overestimation of σ when compared to GPS collar data was more pronounced for the 180-day sessions than the 90-day sessions. We conclude that one-off ('snap-shot') short-term, small-scale camera trap surveys do not sufficiently sample wide-ranging large carnivores. When using SCR models to estimate the density from these data, we caution against the use of poor sampling designs and/or misinterpretation of scope of inference. Although the density estimates from one-off, short-term, small-scale camera trap surveys may be statistically accurate within each short-term sampling period, the variation between density estimates from multiple sessions throughout the year illustrate that the estimates obtained should be carefully interpreted and extrapolated, because different factors, such as temporal stochasticity in behaviour of a few individuals, may have strong repercussions on density estimates. Because of temporal variation in behaviour, reliable density estimates will require larger samples of individuals and spatial recaptures than those

https://www.panthera.org/ NGO played no role in study design, data collection, analysis or decision or preparation of this manuscript. HQ Summerlee Foundation http://summerlee.org/ Foundation played no role in study design, data collection, analysis or decision or preparation of this manuscript.

**Competing interests:** The authors have declared that no competing interests exist.

presented in this study (mean +/- SD = 14.2 +/- 1.2 individuals, 37.7 +/- 4.7 spatial recaptures, N = 276 sessions), which are representative of, or higher than published sample sizes. To satisfy the need for larger samples, camera surveys will need to be more expansive with a higher density of stations. In the absence of this, we advocate longer sampling periods and subsampling through time as a means of understanding and describing stability or variation between density estimates.

## Introduction

Managing wildlife species for conservation or harvest requires accurate estimates of population size, to assess population viability, with enough precision to detect significant change through time or across space [1]. The unit of analyses is population abundance. Researchers are rarely able to measure the abundance of the entire population. Often, logistics limit researchers to sampling a fraction of the area inhabited by the population of interest (survey area). Instead of measuring population size, researchers estimate population density as the number of individuals present within the arbitrarily defined survey area where the sampling occurs [1]. If the density estimates are robust, we can compare them within a survey area through time, between survey areas in space, or extrapolate to the wider landscape to estimate the total population abundance. For robust estimates, survey areas should be large enough and/or have sufficient number of individuals detected with sufficient number of recaptures, minimising the effects of movement in and out of the survey area [2] associated with social or stochastic events. This ensures that we can distinguish between temporal variation in local abundance and true population change.

The difficulty of adequately sampling in space and time for population assessment is most notable for highly mobile, wide-ranging terrestrial species, especially if they live at low density. For these species, surveys must span large areas to sample enough individuals for population estimation. We expect that small-scale sampling will result in high temporal and spatial flux in detection rates, which will only average out if more individuals are sampled at the landscape level. Large forest-dwelling carnivores, such as jaguars (*Panthera onca*) are a good example of a wide-ranging species for which population status is difficult to assess. The advent of camera traps presented a new paradigm for monitoring populations of such elusive carnivores, using photo records within a closed population capture-recapture (CR) framework, or, more recently, a closed population spatially explicit capture-recapture framework (SCR; for a review see [1]). Closed population CR estimators assume closure in sampled populations in space and time, i.e. no individuals enter or leave the survey area during the sampling period [2]. To achieve this, survey periods must be sufficiently short to ensure no births, deaths, immigration or emigration, but long enough to obtain sufficient captures and recaptures for accurate and precise estimation. Following [3], researchers studying large carnivores have almost universally opted for a compromise survey period of two to three months to ensure population closure and sufficient captures. However, most two to three-month (from hereon, 'short-term') camera trap studies of large carnivores, and jaguars in particular, report low samples of individuals, frequently < 10, and rarely > 20 individuals [1, 4]. They move in response to each other and un-sampled individuals present inside and beyond the survey grid, resulting in detection records that vary between repeated surveys, depending on the local conditions at the time of sampling (e.g. males searching for females in heat, young adults dispersing or establishing territories). This local variation is likely independent of population change we intend to

monitor (e.g. [5]). Simulation studies support the use of longer sampling periods for low-density, long-lived species [6]. Sampling for longer increases the number of recaptures, which improves the accuracy and precision of the density estimate, averaging out any temporal variation in the spatial behaviour of the detected individuals.

From a population perspective, we might question the utility of density estimates across arbitrarily chosen small-scale study areas, if they are not representative of the population. By way of a conceptual example, consider a migratory species. If sampled in the short-term and at the small-scale, detection records of the target species might show no detection, some presence, or high abundance, depending on the timing of the study. Independently these measurements are not useful if we cannot place them within the context of the population system that we are studying. If the species migrates predictably in space and time, repeat surveys at the same time each year would allow monitoring of population change but not necessarily of population size. If the migratory behaviour fluctuates in space and time, the researcher must extend the sampling period or sample across the entire landscape.

Published estimates of large carnivore densities from camera trap data frequently represent 'snap-shots' of single study sites [1]. In the absence of knowing whether the method of density estimation is robust across time, the utility of comparing snap-shot estimates between study sites (e.g. [7, 8]), or extrapolating across wider landscapes (e.g. [9]) is questionable. For example, [5, 10] detected variation between abundances and densities of male jaguars estimated from systematic three-month camera surveys, repeated during the same months each year for 12 years. They surmised that the variation was attributable to the combination of the relatively small survey area and the idiosyncrasies of the social situation and status of the sampled individuals during each survey period, rather than due to real change in population size over time at the landscape level.

In this study, we investigate whether density estimates are robust through time, using a well-studied population of protected jaguars in Belize, Central America [5]. We maintained camera traps at the same locations for a year, sub-sampled the photo record data into 276 3-month detection records and 186 6-month detection records, and repeatedly estimated density of the same population through time, using maximum likelihood (ML) spatially explicit capture-recapture (SCR) models [11, 12]. The ML SCR models derive density by simultaneously estimating the capture probability (g0) and a ranging parameter, sigma ($\sigma$), from the spatial distribution of the sampled individuals [11, 12]. The model assumes that each individual has an activity centre where capture probability (g0) is highest, if detectors were placed there. A detection function models the decrease of capture probability with increased distance from the estimated activity centre, with $\sigma$, the scale parameter, representing the rate of decrease (e.g. [11, 12]). The estimated $\sigma$ describes the mean home range use of the detected individuals during the sampling period. Variation between estimates of $\sigma$ will reflect variation in the ranging behaviour of the sampled individuals between sampling periods. Variation in g0 will reflect variation in capture probability, which is directly related to the estimate of abundance within the traditional CR framework.

Accurate estimation of $\sigma$ requires sufficient spatial recaptures to sample movement. Empirical and simulation studies have recommended that survey areas should be $\geq 1$ home range size for unbiased SCR density estimates [4, 13]. As $\sigma$ reflects home range use, we can transform estimated $\sigma$ values into an estimate of the mean home range area, assuming a half-normal detection function and roughly circular ranges [14]. Therefore, we can assess the accuracy of the estimate of $\sigma$ using independent data on home range size of individuals from the study area.

We examine the variation in SCR density estimates and their precision throughout an entire year. We investigate whether the estimates vary with: (1) the ranging behaviour of

individuals, specifically the scale parameter (σ) and the number of spatial recaptures; and (2) the abundance and demographic structure of the local population, specifically the capture probability (g0) and number of individuals and detections of each sex. In this way, we investigate whether the density estimates are sensitive to changes in the ranging behaviour of the sampled individuals and/or idiosyncrasies associated with local demographic changes (represented by the number and detections rate of males and females). We conduct this assessment using two sets of sampling periods, the traditional 3-month period (90 days) [3] and an extended period of 6-months (180 days) [6]. Additionally, we test the accuracy of the SCR estimates of the scale parameter, σ, by comparing home range use obtained from GPS collared individuals within the study area, with circular home range transformations of estimated σ values. This is the first study to measure variation in density estimates of jaguars from the same survey area across an entire year.

## Methods

### Study area

We conducted fieldwork in the Cockscomb Basin Wildlife Sanctuary (here-on, CBWS or sanctuary), in Belize, Central America (Fig 1, 16˚45'50.49"N / 88˚30'18.40"W; projection: GCS

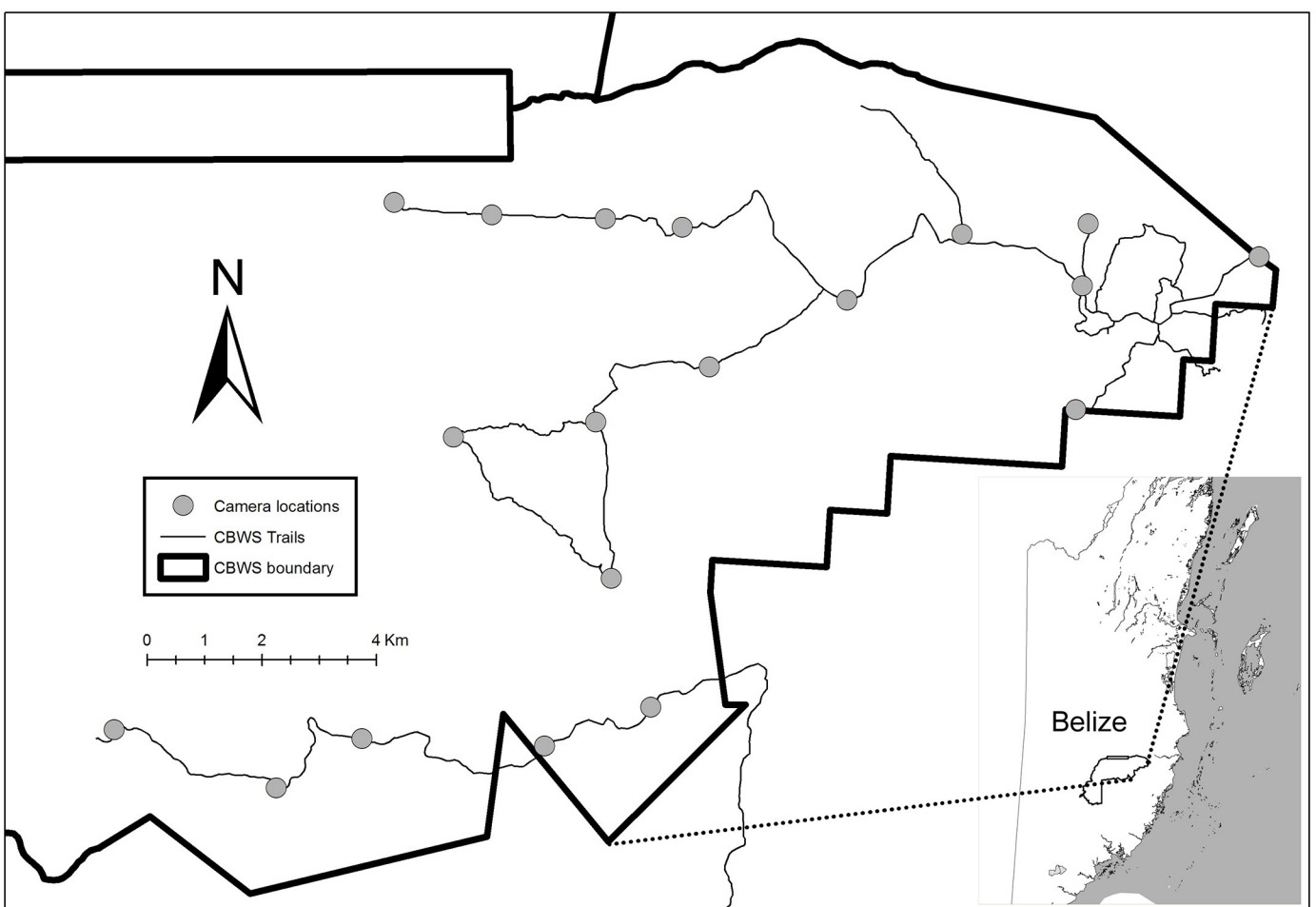

**Fig 1. Study area within the Cockscomb Basin Wildlife Sanctuary (CBWS), showing the 20 permanent camera station locations and the trail system.**

WSG84). The area comprises 490 km$^2$ of moist seasonal broad-leaved tropical forest that was selectively logged until 1981, protected in 1986, and is now a mosaic of regenerating secondary forest in several stages of succession. There is a distinct dry season from February to May with the remainder of the year considered wet season. Indigenous people with traditional knowledge of the local area have reported that the frequency of jaguar calling, associated with jaguar mating, is greatest during the period November to January (personal observation Harmsen & Foster).

Many of the old logging roads in the east of the CBWS are maintained as tourist trails or patrol routes (Fig 1), providing easy, and presumably, preferred travel routes for jaguars to move through the dense secondary vegetation [15, 16]. Compared to other tropical moist broad-leaved sites, the CBWS supports a relatively high density of jaguars [17].

## Camera trap survey

We maintained 20 paired camera stations along the trail system for one year (365 days) from March 2013 to March 2014, covering an area of ~120 km$^2$ (Fig 1). Neighbouring stations were separated by 1.07 to 3.05 km (mean = 2.02 km). The furthest distance between any two stations was 21.6 km. We used white-flash digital camera traps (Pantheracam V3) with a minimum delay of 8 seconds between successive photo triggers. Every photograph was stamped with the time and date. We identified adult individual jaguars only based on their unique spot patterns, and assigned sex based on the presence or absence of testicles, following [18]. We excluded detections of the same individual at the same camera station on the same day. We recognise that sampling only on trails introduces the potential for sampling bias. Although random sampling (including off-trail locations) would be preferable, the capture probability of jaguars at off-trail camera stations falls close to zero, providing insufficient detections for reliable density estimation [18, 19].

## Density estimation

We estimated the jaguar density (D), detection probability (g0) and sigma ($\sigma$) for every 90-day and 180-day period ('session') within the 365-day camera-trap record. Each session comprised 90 or 180 consecutive one-day occasions. Thus, for the 90-day period, the first session ran from day 1 to 90, the second from day 2 to 91, the third from day 3 to 92, and so on, until the final session from day 276 to 365; creating a rolling window of 276 parameter estimates over the year. This was repeated for the 180-day period, having the first session from day 1 to 180, day 2 to 181 etc., until session 186 to 365, creating a rolling window of 186 parameter estimates over the year. We estimated D, g0 and $\sigma$ using maximum likelihood spatially explicit capture-recapture analysis, using the package 'secr' with default settings in R [11, 12, 20]. We used a buffer of 30 km to define the area of interest (mask) based on conservative estimates of home range radius of 9 km [21], with a spacing between adjacent points of 150 to 300 m within the mask [22], and the most commonly-used detection function, half normal. We recorded the number of spatial recaptures per session to check that they met the recommended threshold of at least 20 spatial recaptures for accurate and precise estimation of $\sigma$ [23].

In our study area, male jaguars have larger ranges and higher detection rates on trail-based camera traps than female jaguars, warranting the use of covariates when estimating $\sigma$ and g0 [4,9]. Therefore, to maintain consistency in model selection, reduce complexity and to aid comparability between sessions, we estimated male density only; and held D, g0 and $\sigma$ constant. However, for every session, we also recorded the number of individuals and number of detections of each sex.

## Parameters influencing density estimates

In order to examine the extent to which the density estimates are influenced by temporal, spatial, and demographic parameters we investigated the variation in the 276 and 186 density estimates through time, and tested for linear or curvilinear relationships between male density and: the number of individuals and number of detections of each sex, total number of male spatial recaptures, g0 and $\sigma$. SCR density estimates depend on the number of individuals sampled, how often the individuals are detected, and how far they move within in the survey grid during the sampling period. If the number of sampled individuals stays the same between sessions, but the extents of their ranges change, then we would expect the estimated density to decrease with increases in spatial recaptures, estimated $\sigma$, and derived effective sampling area. If the number of sampled individuals changes between sessions, but the extents of their ranges remain the same, then we would expect the estimated density to increase with the number of detected individuals. In the latter scenario, we would also expect the estimated density to increase with the estimated g0 and the number of detections.

In order to investigate the mechanism driving change in local demographic structure between sessions, we tested for correlations between the mean number of spatial recaptures per male and the following variables: number of male individuals, female individuals, detections of males, detections of females. We assumed the following: (1) if the detected individuals used the trail system more often (more 'active'), they would trigger the cameras more frequently, therefore the number of detections would increase; (2) if the detected individuals moved further along the trail system they would trigger more camera locations; therefore, we would detect more spatial recaptures per individual; (3) if more individuals used the trail system, we would detect more individuals. As a behavioural mechanism to facilitate the search for mates, courtship, and mating, we hypothesised that males would move further during sessions with fewer females and low female activity on the trail system, and their ranges would contract during sessions when more females were more active on the trail system.

## Comparison of σ with GPS derived home range estimates

In order to examine the extent to which σ accurately reflects home range size, we compared our estimates of $\sigma$ from the SCR analysis with estimates of $\sigma$ derived from the known home ranges of GPS collared jaguars in the study area. Assuming a circular home range with a single activity centre and half-normal detection function, [14] estimated approximate home range size as $18.86\sigma^2$. We rearranged this formula as: $\sigma = (\text{home range} / 18.86)^{1/2}$ to estimate the equivalent σ values based on the home range sizes of two male jaguars. Both collared individuals were detected by our camera traps throughout this survey period, and GPS tracking began in 2015 (Harmsen unpublished data). Male 1 was at least five years old and in good body condition and tracked for 348 days. Male 2 was at least nine years old and was tracked for 202 days. For each jaguar, we calculated the 100% minimum convex polygon (MCP) for the entire period of tracking, and ranges for equivalent 90-day periods, using a similar shifting set of 90-day sessions as for the density estimates (day 1 to 90, day 2 to 91, day 3–92 and so on), giving 258 shifting sessions for Male 1 and 202 shifting sessions for Male 2. For all MCPs we estimated the equivalent σ values using the formula above. The validity of using sigma as a measure of home range size relies on the detection process mirroring the half-normal detection function [14]. We recognise the discrepancy in a direct comparison with 100% MCPs derived from GPS collar data. We also estimated the 95% kernels from the collar data, but they were generally smaller than the 100% MCPs, therefore we present the 100% MCPs only. The 100% MCP is the most standardised and easy to interpret home range estimator, as well as

accurate and precise, when considering the high rate of daily location samples from the GPS collars.

As our sample of collared individuals was low (N = 2 individuals), we assessed whether the ranges of the two collared males were representative of the sample of males detected by the camera traps. We did this by comparing the maximum distance between photo recaptures for every male with spatial recaptures, including the collared males, across the study period (365 days) and then ranked the males by their maximum distance moved.

## Results

During one year of continuous monitoring with camera traps, we detected 21 male jaguars, with a mean of 14 males (range 13 to 17) and 140 detections (range 113 to 170) per 90-day session (N = 276 sessions) and a mean of 17 males (range 14 to 19) and 281 detections (range 249 to 306) per 180-day session (N = 186). During the same period, we detected 12 female jaguars, with a mean of 6 females (range 3 to 7) and 18 detections (range 7 to 31) per 90-day session (N = 276 sessions) and mean of 9 females (range 7 to 10) and 35 detections (range 24 to 44) per 180-day session (N = 186).

### Density estimates

Across the 276 90-day sessions, mean density estimates of males ranged from 1.11 to 2.98 individuals per 100 km$^2$ (mean = 2.01, N = 276 sessions); with the minimum lower and maximum upper 95% confidence intervals ranging from 0.51 to 5.30 male jaguars per 100 km$^2$; and a mean precision per estimate of 2.45 male jaguars per 100 km$^2$ (range 1.64 to 3.86; Fig 2—left panel). For the 186 180-day sessions, density estimates were similar but more precise than for the 90-day sessions. For the 186 180-day sessions, mean density estimates of males ranged from 1.27 to 2.28 individuals per 100 km$^2$ (mean = 1.79, N = 186 sessions); with the minimum

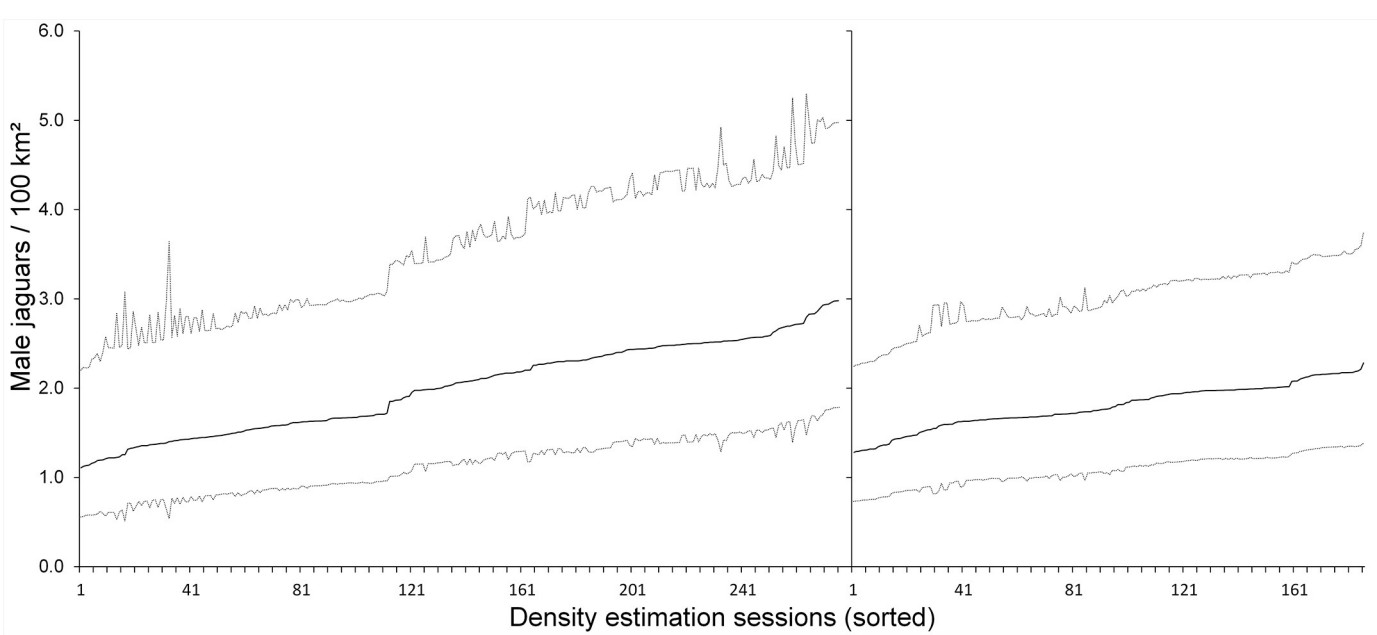

**Fig 2. SCR density estimates of male jaguars ordered by mean estimated density from low to high for 276 90-day samples (left panel) and 186 180-day samples (right panel) sampled from 365 days of continuous monitoring with 20 camera trap stations in Belize.**

lower and maximum upper 95% confidence intervals ranging from 0.73 to 3.75 male jaguars per 100 $km^2$; and a mean precision per estimate of 1.92 male jaguars per 100 $km^2$ (range 1.51 to 2.37; Fig 2—right panel).

## Parameters influencing density estimates

**Time series.**   For the 90-day sessions, density oscillated approximately on a three-monthly cycle, with mean density, per 100 $km^2$, initially increasing from 1.53 (session 1) to 2.90 (session 110) then dropping back to 1.21 (session 175) and rising again to 2.98 (session 276) (Fig 3a—left panel). Within this broad pattern were some abrupt changes between consecutive sessions, for example, between session 201 and 202, the density estimate almost doubled from 1.36 to 2.64 male jaguars per 100 $km^2$. In this instance, the removal of one occasion (day 201, 5-Oct-2013) resulted in the loss of one individual from the dataset which had a high detection rate at a single location in previous sessions, while the newly added day (day 291, 3-Jan-2014) resulted in the addition of three new detections of two individuals already detected in session 201.

For the 180-day sessions, the oscillations in density were less pronounced than the 90-day sessions, but of roughly similar shape, with mean density per 100 $km^2$ initially increasing from 1.28 (session 1) to 2.08 (session 77) then dropping back to 1.68 (session 94) and rising again to 1.99 (session 186) (Fig 3a—right panel). Abrupt changes in density estimate between sessions were of lower magnitude compared to the 90-day sessions: from session 128 to 129, the density increased by almost one-half, from 1.60 to 2.28.

**Capture probability (g0).**   For the 90-day sessions, the mean g0 was low but varied up to three-fold over the 276 sessions (mean = 0.06, range: 0.03 to 0.1, N = 276, Fig 3b—left panel). The variation was not erratic, increasing gradually from session 1(Mar-Jun-2013) onwards, then plateauing from session 162 (Aug-Nov 2013) until session 226 (Oct 2013 –Jan-2014), then declining to its former level. The precision of g0 decreased during the plateau (maximum g0 CI range = 0.25). We found no association between g0 and male density, the number of female individuals or the number of female detections. However, g0 increased with the number of male detections (Pearson correlation $r = 0.79$, $p < 0.01$, N = 276), while showing no relationship with the number male individuals. During the plateau period of high capture probability (sessions 162 to 226), we detected few males (13–14) and male density was low, but those detected had a high capture probability and relatively few spatial recaptures (Fig 3a and 3b—left panel, Fig 4a, 4b and 4c—left panels).

Compared to the 90-day sessions, the mean g0 for the 180-day sessions was lower and varied only up to twofold (mean = 0.05, range: 0.03 to 0.07, N = 186, Fig 3b-right panel) with a less pronounced plateau, (sessions 121 to 134) and greater precision. As with the 90-day sessions, g0 did not vary with male density or the number of male individuals but increased with the number of male detections (Pearson correlation $r = 0.89$, $p < 0.01$, N = 186). Unlike the 90-day sessions, g0 also increased with the number of female individuals and the number of female detections (female individuals: Pearson correlation $r = 0.67$, $p < 0.01$, N = 186; female detections: Pearson correlation $r = 0.79$, $p < 0.01$, N = 186; Fig 3b—right panel, Fig 4b and 4c —right panels), while decreasing with the mean number of spatial recaptures per male (Pearson correlation $r = -0.73$, $p < 0.01$, N = 186; Fig 3b—right panel, Fig 4a—right panel).

**Sigma (σ) and spatial recaptures.**   For both the 90-day and 180-day sessions, increases in σ were associated with decreases in density, and vice versa (Fig 3 and 3c). We detected a strong inverse relationship between the σ estimates and the density estimates (90-day sessions: linear $R^2 = 0.80$, p < 0.01; curvilinear $R^2 = 0.81$, $p < 0.01$; N = 276; 180-day sessions: linear $R^2 = 0.86$, p < 0.01; curvilinear $R^2 = 0.83$, $p < 0.01$; N = 186; Figs 3a, 3c and 5).

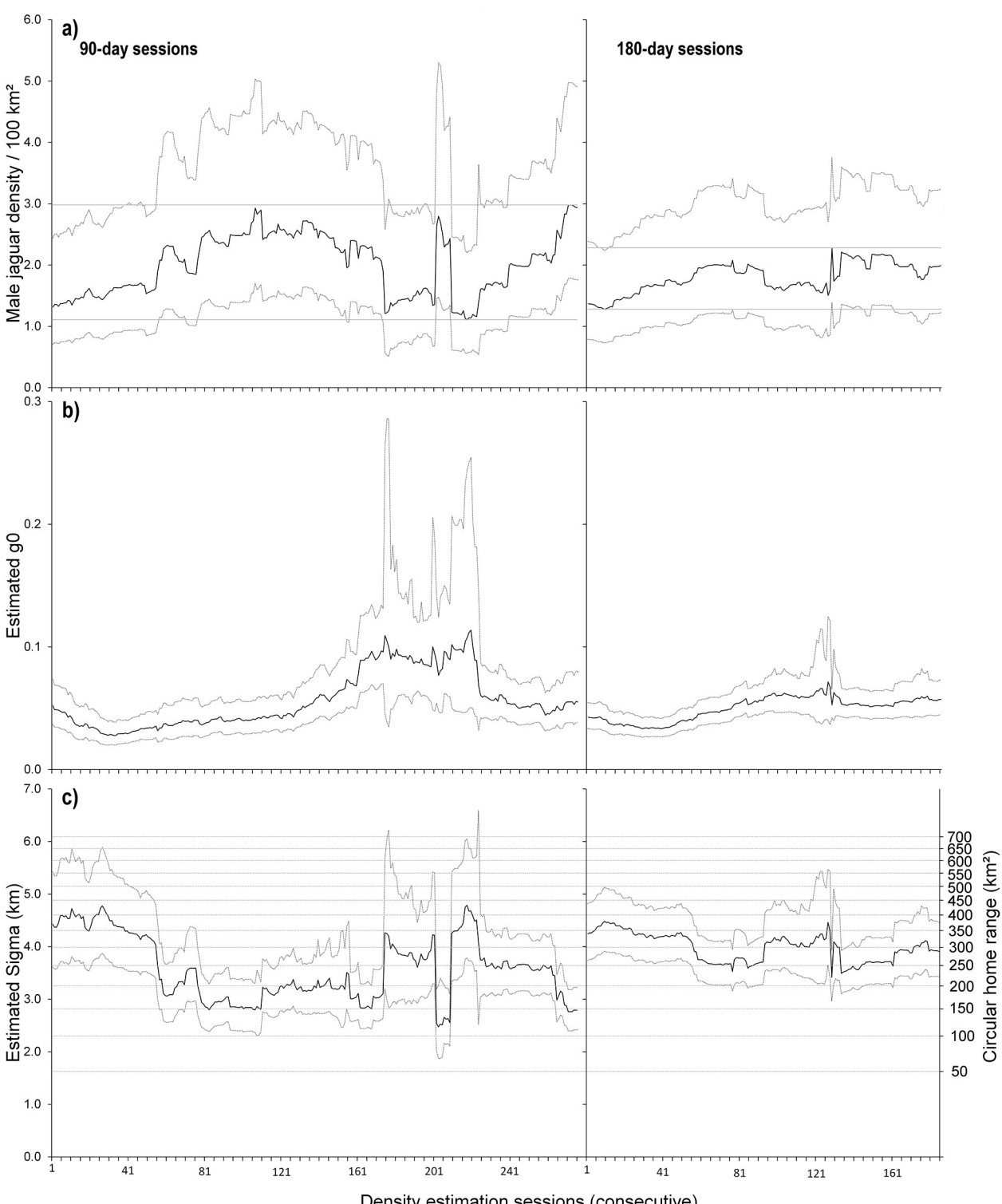

**Fig 3. SCR estimates of density, capture probability (g0) and sigma (σ) of male jaguars for 276 90-day (left panel) and 186 180-day samples presented as consecutive sessions sampled from 365 days of continuous monitoring with 20 camera trap stations in Belize.** For the 90 days, session 1 represents 19-Mar to 16-Jun-2013 and session 276 represents 19-Dec-2013 to 18-Mar-2014; for the 180 days, session 1 represents 19-Mar to 14-Sep-2013 and session 276 represents 20-Sep-2013 to 18-Mar-2014 **a)** Mean estimated density (male jaguars/100km²; black line) with upper and lower confidence interval (dashed line). The horizontal lines show the highest and lower lowest of the mean density estimates. **b)** Mean estimated capture probability at the activity centre (g0, black line) with upper and lower confidence interval (dashed line). **c)** Mean estimated σ (km; black line) with upper and lower confidence interval (dashed line). Dashed grey lines show equivalent values of σ as circular home ranges (km²) on right-hand y-axis.

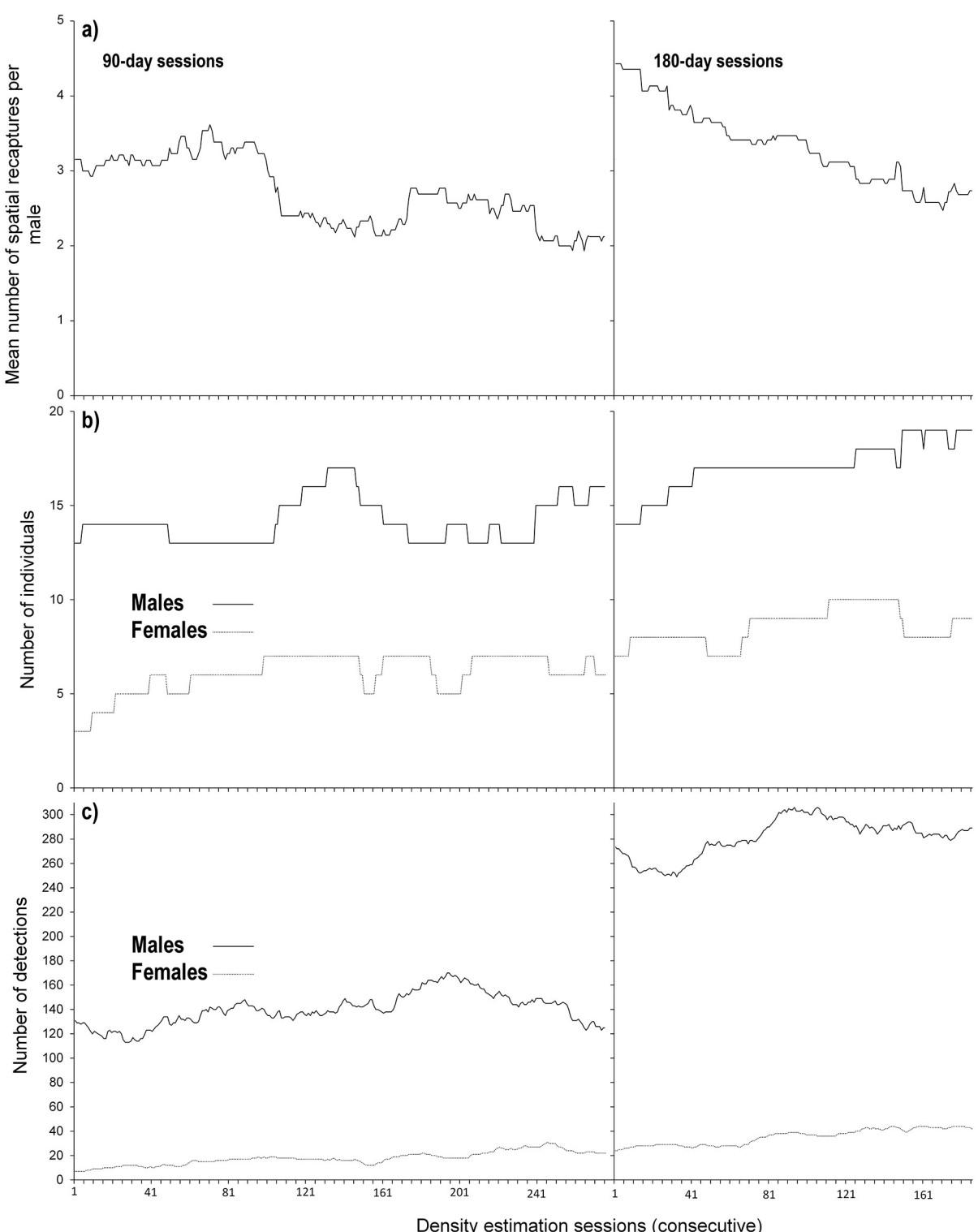

**Fig 4. Jaguar detections for a total of 276 90-day (left panel) and 186 180-day (right panel) consecutive sessions sampled from 365 days of continuous monitoring with 20 camera trap stations in Belize.** For the 90 days, session 1 represents 19-Mar to 16-Jun-2013 and session 276 represents 19-Dec-2013 to 18-Mar-2014; for the 180 days, session 1 represents 19-Mar to 14-Sep-2013 and session 276 represents 20-Sep-2013 to 18-Mar-2014 **a)** Mean number of spatial recaptures per male, per session. **b)** Number of individuals per session (solid line = males, dashed line = females) **c)** Number of independent captures per session (solid line = males, dashed line = females).

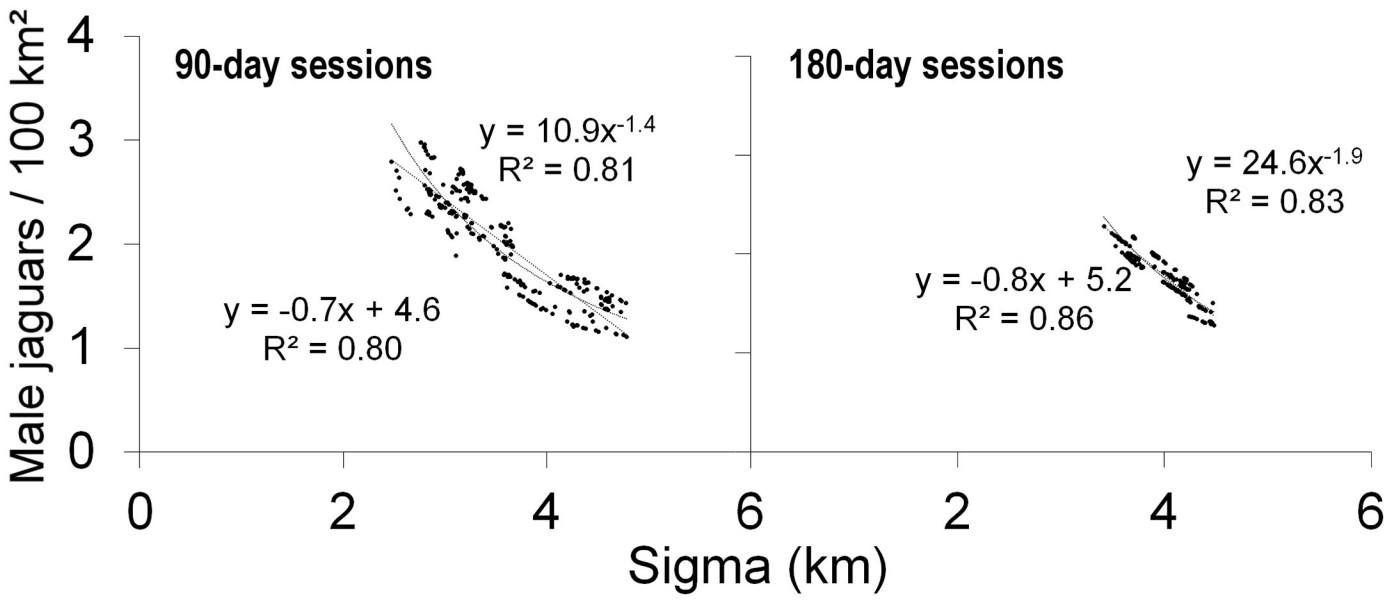

**Fig 5. Regression of male jaguar density against sigma (σ) for 276 90-day (left panel) and 186 180-day sessions (right panel), showing linear and curvilinear regressions.**

The total number of male spatial recaptures per session ranged from 29 to 47 for the 90-day sessions (mean ±SD = 37.7 ± 4.7 spatial recaptures) and from 47 to 63 for the 180-day sessions (56.1 ± 4.6 spatial recaptures). We found no relationship between density and the mean number of male spatial recaptures for the 90-day sessions, while for the 180-day sessions density decreased with an increase in mean male spatial recaptures (Pearson correlation $r$ = -0.75, $p < 0.01$, N = 186; Fig 3a—right panel, Fig 4a—right panel), suggesting a negative relation between range expansion of individual jaguars and density.

**Abundance.** For the 90-day sessions, we found no evidence that male density varied with the number of male individuals or male detections (Fig 3a—left panel, Fig 4b and 4c—left panel). However, across the first 160 consecutive 90-day sessions (the 250 days prior to the plateau in g0), male density increased with both the number of female detections (Pearson correlation, $r$ = 0.90, $p < 0.01$, N = 160 sessions, Fig 3a—left panel, Fig 4c—left panel, Fig 6—left panel), and the number of female individuals (Pearson correlation, $r$ = 0.85, $p < 0.01$, N = 160 sessions; Fig 3a—left panel, Fig 4b—left panel, Fig 6—right panel). These relationships broke down beyond the first 250 days of the survey (160th 90-day session, Fig 4b and 4c left panel, Fig 6).

For the 180-day sessions, we found no relation between male density and male detections, but male density increased with the number of male individuals (Pearson correlation, $r$ = 0.81, $p < 0.01$, N = 186 sessions, Fig 3a—right panel, Fig 4b—right panel). The relationship was stronger for the first 160 consecutive sessions (Pearson correlation, $r$ = 0.83, $p < 0.01$, N = 160 sessions, Fig 3a—right panel, Fig 4b—right panel). We found no evidence that male density varied with the number of female individuals or number of female detections for the 180-day sessions.

**Demographic structure.** For both the 90-day and 180-day sessions, the mean number of spatial recaptures per male decreased with the number of female detections (90-days: Pearson correlation $r$ = -0.59, $p < 0.01$, N = 276 sessions; 180-days: $r$ = -0.89, $p < 0.01$, N = 186 sessions; Fig 4a and 4c), and the number of female individuals (90-days: Pearson correlation $r$ = -0.48, $p < 0.01$, N = 276 sessions; 180-days: $r$ = -0.49, $p < 0.01$, N = 186 sessions; Fig 4a and 4b),

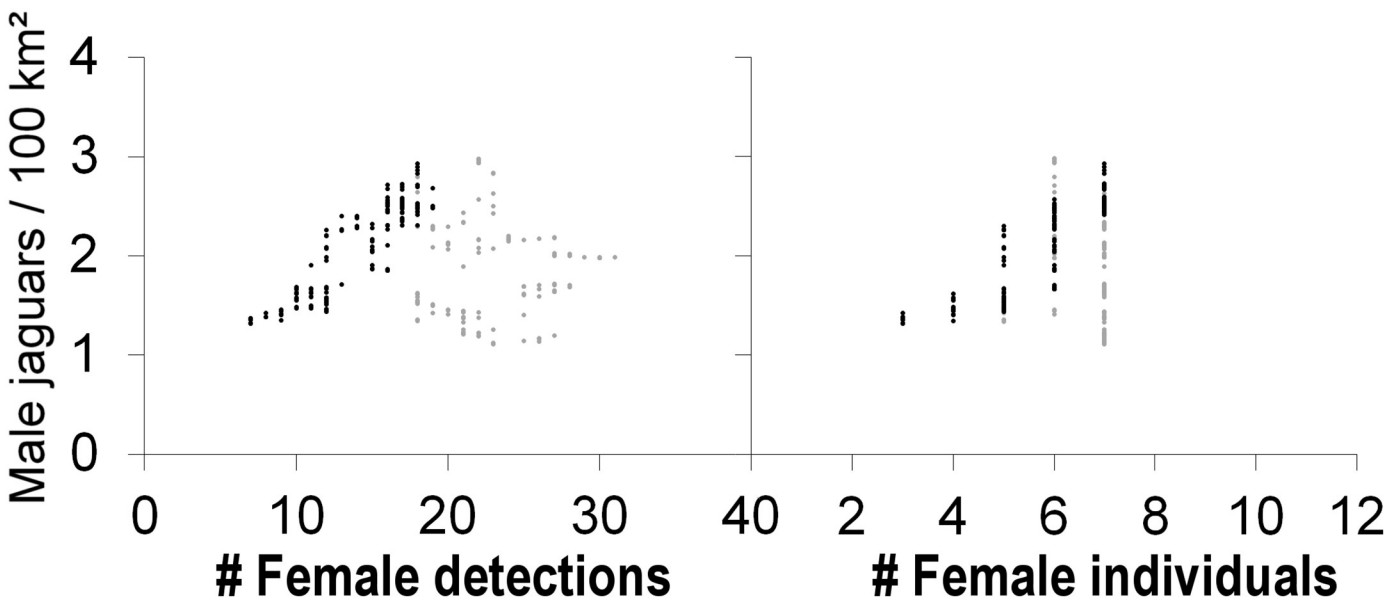

**Fig 6. Variation in SCR estimates of male jaguar density with female detections, and female individuals for 276 90-day sessions.** Left panel: female detections (grey dots for the total series, and black dots for the subset sessions 1–160). Right panel: female individuals (grey dots for the total series, and black dots for the subset of sessions 1–160).

indicating that increased space use by males was associated with a decreased detection of females, and vice versa.

The mean number of male spatial recaptures per male also decreased with the number of male individuals (90-days: Pearson correlation $r$ = -0.69, $p$ < 0.01, N = 276 sessions; 180-days: $r$ = -0.96, $p$ < 0.01, N = 186 sessions; Fig 4a and 4b), and the number of male detections for the 180-day session only (Pearson correlation $r$ = -0.64, $p$ < 0.01, N = 186 sessions; Fig 4a and 4c —right panels), suggesting that with decreased number of males, ranges increase and as males range further they are less frequently detected by the camera traps.

**Comparison of σ with GPS derived home range estimates.** The maximum distance moved by a male jaguar between camera stations was 18.6 km. For the 17 male jaguars with spatial recaptures, including the two collared males, the mean maximum distance moved across the 12-month period was 8.4 (SD ± 5.2) km. The two collared males ranked 2nd and 3rd highest in the maximum distance moved (male 1: 17.7 km, male 2: 12.4 km), indicating that they were among the wider-ranging individuals sampled within our survey grid.

The 276 estimates of σ, the scale parameter derived from the SCR analysis for the 90-day sessions, were larger than the associated back-transformed 100% MCP ranges of two GPS collared male jaguars in the study area (Table 1, Fig 3c). All of the SCR estimates of σ were ≥ 2.5km, and 93% were >2.8km, the largest estimate from the GPS data, suggesting that the SCR model overestimates σ when compared with empirical telemetry data, even if we use 100% MCP (Table 1, Fig 3c). All 186 estimates of σ from the SCR analysis for the 180-day sessions were > 2.8 km (mean (± SD) = 4.0 (±0.6), range = 3.4 to 4.5 km), including the lowest confidence interval (3 km).

## Discussion

Our 276 estimates of male jaguar density, using 90-day sessions across a year, ranged three-fold from 1 to 3 males per 100 km², with a ten-fold range between the lower-most and upper-

**Table 1. Home range estimates from GPS collar locations for two male jaguars in the Cockscomb Basin Wildlife Sanctuary, Belize (Harmsen unpubl. data) with back transformations to sigma (σ) based on a circular home range of the same size with a single activity centre (following [14]); and SCR estimates of sigma (σ) of male jaguars for 276 90-day samples from 365 days of continuous monitoring with 20 camera trap stations in Belize, showing back transformations to circular home range area.**

| | Tracking | 100% MCP (km$^2$) | | | | Equivalent σ (km) | | | |
| | period | All locations | 90-day sessions | | | All locations | 90-day sessions | | |
| | (days) | | Mean (SD) | Range | N sessions | | Mean (SD) | Range | N sessions |
|---|---|---|---|---|---|---|---|---|---|
| **Male 1** | 348 | 150 | 100 (7) | 89–116 | 258 | 2.8 | 2.3 (0.1) | 2.2–2.5 | 258 |
| **Male 2** | 202 | 114 | 85 (11) | 68–100 | 112 | 2.5 | 2.1 (0.1) | 1.9–2.3 | 112 |
| σ | 90 | N/A | 248 (85) | 116–432 | 276 | N/A | 3.6 (0.6) | 2.5–4.8 | 276 |

most confidence intervals (0.5 to 5 males per 100 km$^2$). In some instances, shifting the 3-month survey period in time by a single 24-h occasion was sufficient to cause a doubling of the density estimate. The cause of variation requires investigation and raises questions about the use of density estimates from a single 3-month survey for comparing within and/or between study sites, or for extrapolation to the wider landscape and beyond. In comparison, our 186 density estimates from the 180-day sessions, a sampling period which is not traditionally used, were more stable across the year, ranging from 1 to 2 males per 100 km$^2$, and more precise, (95CI 0.7 to 3.8). This finding supports SCR simulation studies by [6] who recommended extending the survey periods for increasing the precision of density estimates of long-lived species. Although the longer survey periods gave more precise and robust density estimates, the scale parameter σ, and thus density (we infer), suffered from lower accuracy compared to the shorter survey periods. Because of the high variation between our density estimates through the year, we recommend that density estimates of low-density, wide-ranging species should be carefully interpreted and extrapolated if derived from short-term, small-scale camera trap surveys.

Within the SCR framework, density is estimated simultaneously with σ, the scale parameter which reflects mean home range use, and g0, the mean capture probability at the activity centre [11,14]. In this study, we found no relationship between density estimates and estimates of g0, but we did find a significant inverse linear relationship between estimates of density and σ, as found by [14]. We infer that a high estimate of σ was associated with a low estimate of density, and vice versa. Comparison of our σ estimates with known home range use of GPS-collared jaguars in the study area indicates that σ was mostly over-estimated in this study, and this was more pronounced for the longer sessions. Although we compare our σ estimates with home ranges derived from only two GPS-collared males, we know that they were among the most wide-ranging individuals sampled within the survey grid, suggesting our estimates of home range (and thus σ) were not negatively biased by the low sample size. SCR models may overestimate σ if the survey grid is so small that individuals are detected with equal probability throughout the entire study area [13]. However, the survey grid need only cover an area the size of an average home range of the target species for the model to perform well, according to simulation studies [4,13]. Our camera grid covered ~120 km$^2$, equivalent or larger than the mean home range of male jaguars in our study area (unpublished data Harmsen). Additionally, none of the detected males were detected at all stations, indicating the grid was large enough to show variation in spatial detections between individuals (67% of the individuals each only occupied ≤ 5 of the 20 stations, and only 4 individuals occupied > 10 stations, maximum of 13, during the 365-day survey). Potentially, the spatial recaptures were too few and/or ranging patterns too variable between individuals and across time, to estimate σ accurately. However, lengthening the survey periods increased the SCR estimates of σ above known values

derived from GPS collared male jaguars in the area. Empirical evidence suggests that closed population SCR estimates of σ are sensitive to heterogeneity in ranging behaviour, resulting in negatively biased SCR density estimates [24]. Using jaguar camera trap data from 12 annual 3-month camera trap surveys, [5,10] found that estimates of jaguar abundance and density from non-spatial robust design open population models and SCR closed population models respectively, fluctuated widely between the years. However, using the same data, but including movement in activity centres within an open population SCR model, [25] estimated density to be stable across the same 12 surveys, and simulation studies showed that the estimated density for each year was negatively biased when this movement is not accounted for. Like other statistical models, SCR analysis assumes a degree of homogeneity across measured population parameters. In this study, the model assumes a circular home range with a single activity centre and half-normal detection function. This is realistic at a spatial scale for which the average home range size forms only a fraction of the study area e.g. [26]. However, at the spatial scale commonly used in published research on large cats (study areas are generally ≤2x the average home range area [4, 13]), heterogeneity in ranging behaviour in time will be expected among the few detected individuals, resulting in considerable variation in estimated parameter values, between sessions. Simulation studies, like [4, 13], do not account for such heterogeneity.

The incorporation of covariates may help when modelling σ for a population with highly heterogeneous ranging patterns. Sex is a covariate easily derived from photo records. However, non-visual/behavioural covariates based on social status or age which may influence ranging behaviour (e.g. dominant/subordinate, resident/ transient, healthy/sick), are difficult to infer from photo records without long-term behavioural study [5]. We may also question the use of the half-normal detection function to model the decay in activity with distance from the activity centres. Although the half-normal detection function is frequently used in SCR models of large carnivores [27–29], there is no good reason to assume that this is a realistic representation of their movement patterns, especially within a 3-month period. If they are better represented by a hazard function, and ranges are overlapping, then one may expect equal probability of detection throughout the home ranges, despite the survey area being the size of an average home range. Because camera stations were restricted to trails only, the detection process sampled the way jaguars move on trails. In contrast, the GPS collars sampled all movement. Where available GPS data may provide more realistic estimates of sigma for density estimation (e.g. [30]). Alternatively, if jaguars use specific environmental features, like rivers and streams, to define their home range distribution, non-Euclidean distance models may be appropriate [31]. It may be difficult to estimate σ reliably using a fixed detection function for a population in which ranging patterns are highly heterogeneous between individuals; especially if the sample size of detected individuals is low, even if the number of spatial recaptures is relatively high.

In this study, we infer that the variation between density estimates is closely associated with variation in the estimated ranging behaviour of individuals between the sessions: variation between range sizes and shapes of detected individuals through time could lead to variation between estimates of σ. This may be further confounded by variation in the best-fitting detection function between sessions. The 'average' ranging behaviour (thus σ) will vary from one session to the next if: (#1) there are insufficient spatial recaptures to reliably estimate σ; and/or (#2) we do not consistently detect the same set of individuals from one session to the next and range use varies significantly between these individuals (e.g. with social status/age); and/or (#3) we detect the same set of individuals from one session to the next, but their range use varies between sessions due to temporal variation in biotic and abiotic factors that influence their movement patterns (e.g. inter and intraspecific interactions, weather conditions. In this study, #1 seems unlikely, as the number of spatial recaptures exceeded 20 for each sample/session

[23]. In the case of #2, we noted that shifting the survey period by a single 24-hour period led to the loss of one individual from the capture record (of 14 individuals) and subsequent doubling of the density estimate. In this case, if the population has been studied long enough to assign social status and/or age of individuals and the sample size is large enough, covariates could be used to model variation in ranging behaviour. However, in the case of #3, individuals' movement patterns vary through time with any number of extraneous factors, so controlling for this between repeat surveys may be impossible. We can improve our understanding of the relationship between estimates of density and σ with respect to behavioural interactions between the sampled individuals by using short but sequential survey periods, as in this study. The shorter the survey period, the more we can assume that the sampled individuals detected in the same locality, detect and influence one another (e.g. [32, 33]). Therefore, while the 90-day survey periods allow us to investigate seasonal or behavioural effects, this is not possible for the longer (180-day) survey periods.

For the 90-day sessions, at the start of the study, male density increased with the number and detection rate of females. When females were rarely present on the trail system, males displayed a wide search pattern (high rate of spatial recaptures), and when female presence increased, males contracted their ranges (low rate of spatial recaptures with high number of detections). As the wet season progressed, we detected fewer males but at a higher rate, and obtained lower estimates of male density, than earlier in the year. This period of fewer but more active males with contracted ranges coincided with the peak period in the number of females and their detection rate. This period spans November to January, noted by local people as when jaguar calling is most frequent (personal observation Harmsen & Foster). We explain these observations in terms of seasonal mating, when receptive females move onto the trail system, becoming temporarily available. As female jaguars are only in heat for 6–17 days [34], males have a short window of opportunity to find and mate with receptive females. Trails are the location for communication between jaguars (spraying, rolling and scraping; [32, 33], ideal for meeting and mating. Males are attracted to the females. During the peak of mating, we assume that subordinate males leave the trail system, and a few dominant competitors remain, monopolising the mating opportunities. Our hypothesis of seasonal changes in mating activity on trails is partially corroborated by the detection of two mating events (male-female pairs displaying courtship behaviour, Fig 7) and two male-female associations (males and females <15 minutes apart at the same locations) during the period of high female activity and low male density, and no mating or close male-female associations outside of this period. During this 'mating' period, the number of males detected decreased and stabilised at 13 to 14 individuals. Similarly, [5] showed that across twelve 3-month surveys conducted annually, the number of permanent resident male jaguars detected by the same camera grid was stable around 13–14 individuals, with a fluctuating layer of transient males. Potentially this represents the resident male carrying capacity. We conclude that the variation between density estimates from the 90-day sessions is better explained by behavioural variation associated with stochastic periods of mating and non-mating than by a real change in population size. Similarly, [35] detected variation between four density estimates from four 3-month surveys conducted over the course of the year in the Llanos of the Venezuela and attributed the peak in density to seasonal mating rather than to a change in population size. We observed less variation between density estimates from the 180-day (6-month) sessions than for the 90-day sessions, presumably because the 'mating' period fell within all the 186 180-day sessions, with every session spanning a complete social cycle (periods of mating and non-mating).

Using sequential surveys, of the size and duration commonly recommended in the literature, and shifting the survey periods by one day at a time over the course of an entire year, we have demonstrated that SCR density estimates of low density, wide-ranging carnivores

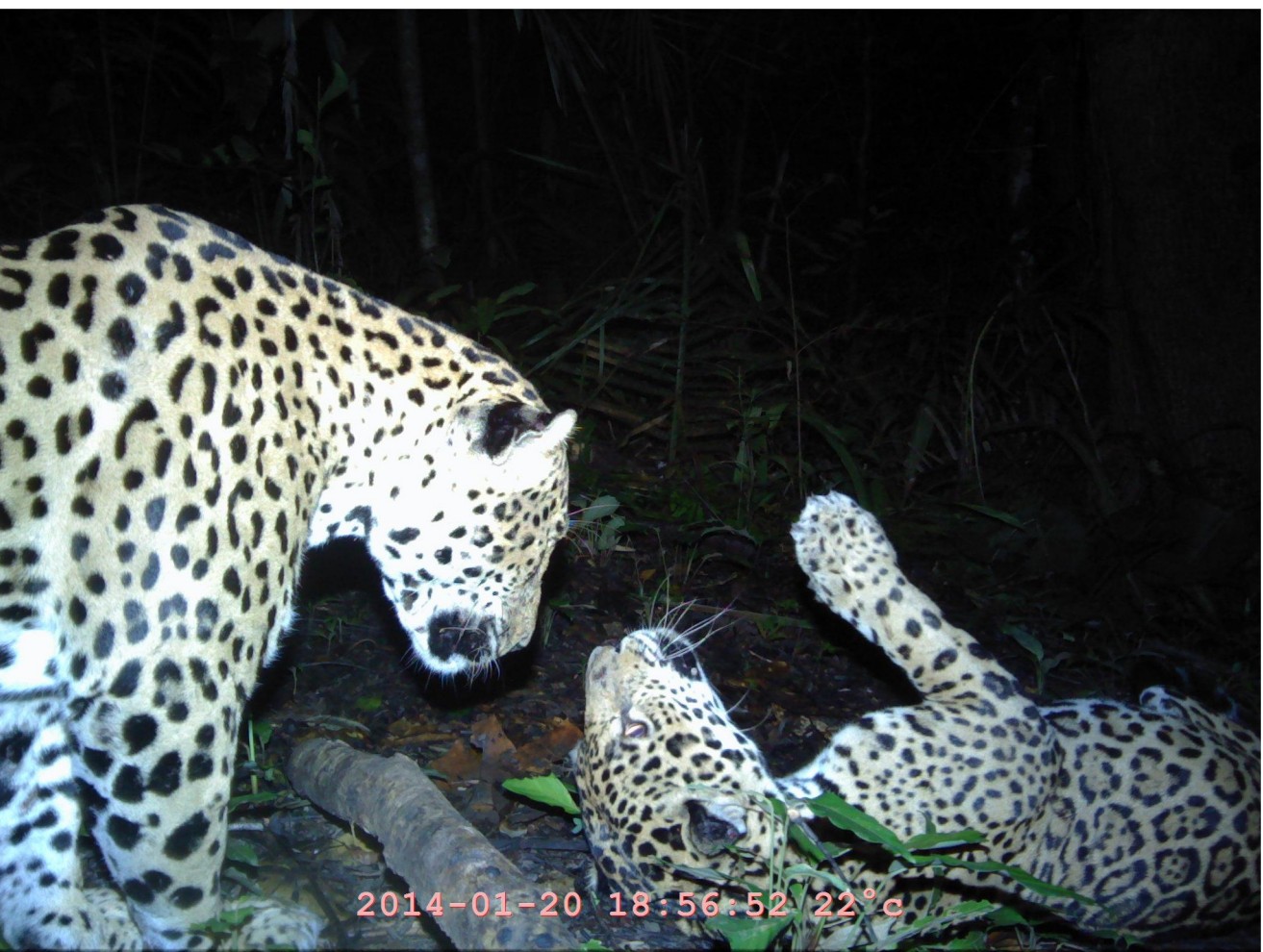

**Fig 7. Camera trap photo of courtship behaviour between male and female jaguar in the Cockscomb Basin Wildlife Sanctuary, Belize in Jan-2014.**

fluctuate through time. Assuming that sample sizes are sufficient to capture a range of individuals' movements, density estimates would only be identical year-round on a static trapping array if the sampled population was demographically closed and either experienced no temporal variation in space use, or sampling accommodated any strong temporal violations of geographic closure or heterogeneous use of space across the year. Therefore, we would not expect most sampling designs to be robust across the year as they must balance logistical feasibility with the assumption of closed periods based on known biology and seasonal space use of the species. However, for most studies of large carnivores, at logistically feasible survey extents, movement in and out of the grid will be stochastic and unpredictable resulting in density estimates that may oscillate between survey sessions, as demonstrated in this study. Because SCR is estimating the instantaneous local density, we recommend that researchers take care in interpreting the scope of inference, considering the estimates within the context of the local demographic, environmental or climatic conditions. Our results bring into question the utility of density estimates of wide-ranging carnivores from single 'snap-shot' surveys, as applied to the assessment of population status. There are at least 131 estimates of jaguar density in the

published and grey literature, from surveys conducted from 2002 to 2014 [30]. The 131 estimates originate from 93 study sites across 15 countries. Over one-half (70/131) of the estimates are from 'one-off' or 'snap-shot' surveys (study site sampled once only). If such point estimates are derived from relatively small survey areas with low sample sizes, we may question what they represent in space and their potential stability through time. We raise concern, therefore, about their recent use in meta-analyses to estimate the current global jaguar range and population [36, 37], particularly as the age of the density estimates used varies, with the oldest study predating the meta-analyses by up to 16 years.

The challenge of obtaining reliable measures of population size of jaguars, or other low density, wide-ranging large carnivores, from camera trap data requires a paradigm shift. If the use of 'snap-shot' surveys (one-off surveys of three-month duration) is to continue, then researchers must increase sample sizes (number of individuals detected and number of spatial recaptures), so that the fluctuation between surveys is insignificant. As shown in this study, the number of individuals detected varied by up to one third between survey sessions (90-day sessions 13 to 17 males; 180-day sessions 14 to 19 males). By sampling more individuals, by using larger survey areas and/or denser camera grids, the sampling sessions may become more robust to temporal variation in unpredictable carnivore behaviour. However, logistical constraints have long limited camera trap surveys, with most camera studies of jaguars and other large carnivores failing to sample more than 10 individuals [1, 4]. Even with extensive and intensive spatial sampling, resulting in higher sample sizes, it would be impossible to assess the level of confidence in estimates from short-term snap-shot surveys in the absence of validation. We should therefore increase the length of the sampling period and subsample, as in this study, to assess temporal variation in density as a measure of confidence in the estimated SCR parameters. We recommend that it becomes standard for researchers to extend survey periods so that they can subsample through time as a means of understanding and describing stability or variation between multiple density estimates from their field sites. The use of recently developed open population SCR models can also be used to understand population change [25]. We recommend the use of empirical camera trap data to capture the complexity inherent in the population dynamics of carnivores and the landscapes they inhabit.

## Supporting information

**S1 Data.**
(CSV)

**S2 Data.**
(CSV)

## Acknowledgments

We thank the Government of Belize, Belize Forest Department, and Belize Audubon Society for providing logistical support for fieldwork. We thank all the staff of the Belize Audubon Society, both field and office, in particular Nicacio Coc and Dominique Lizama for their help over the years. We thank Emma Sanchez and Vivian Soriero for additional field assistance and data organisation, and Rebecca Wooldridge, Yahaira Urbina, Sarah Elkin, and Nicola Saville for assistance with running models. We thank Prof. C. P. Doncaster for his continued support over the years. We thank the Panthera Conservation Science team and two anonymous reviewers who helped improve the manuscript considerably. This paper is dedicated to the memory of Dr. Alan Rabinowitz, for his inspirational work initiating jaguar research in the Cockscomb Basin, and his support and guidance over the years.

## Author Contributions

**Conceptualization:** Bart J. Harmsen, Rebecca J. Foster, Howard Quigley.

**Data curation:** Rebecca J. Foster.

**Formal analysis:** Bart J. Harmsen.

**Funding acquisition:** Rebecca J. Foster, Howard Quigley.

**Investigation:** Bart J. Harmsen.

**Methodology:** Bart J. Harmsen, Howard Quigley.

**Project administration:** Bart J. Harmsen, Rebecca J. Foster, Howard Quigley.

**Supervision:** Rebecca J. Foster, Howard Quigley.

**Writing – original draft:** Bart J. Harmsen.

**Writing – review & editing:** Bart J. Harmsen, Rebecca J. Foster, Howard Quigley.

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
