## [Decision Letter · Decision Letter 0]

11 Feb 2020

PONE-D-19-35092

Spatially explicit capture recapture density estimates: robustness, accuracy and precision in a long-term study of jaguars (Panthera onca)

PLOS ONE

Dear Dr. Harmsen,

Thank you for submitting your manuscript to PLOS ONE. After careful consideration, we feel that it has merit but does not fully meet PLOS ONE’s publication criteria as it currently stands. Therefore, we invite you to submit a revised version of the manuscript that addresses the points raised during the review process.

Both reviewers enjoyed the topic of your manuscript. However, it could benefit from rewording to ease the reading and to make clearer your hypothesis and conclusion. Perhaps to provide some different ways to circumvent the raised issue would be helpful.

Please comment/follow the suggestion of both reviewers.

We would appreciate receiving your revised manuscript by Mar 27 2020 11:59PM. To enhance the reproducibility of your results, we recommend that if applicable you deposit your laboratory protocols in protocols.io, where a protocol can be assigned its own identifier (DOI) such that it can be cited independently in the future. For instructions see: http://journals.plos.org/plosone/s/submission-guidelines#loc-laboratory-protocols

We look forward to receiving your revised manuscript.

Kind regards,

Guillaume Souchay

Academic Editor

PLOS ONE

Journal Requirements:

1. In your Methods section, please provide additional location information of the study area, including geographic coordinates for the data set if available.

2. Our internal editors have looked over your manuscript and determined that it is within the scope of our Biodiversity Conservation Call for Papers. This collection of papers is headed by a team of Guest Editors for PLOS ONE (https://collections.plos.org/s/biodiversity). The Collection will encompass a diverse range of research articles on biodiversity conservation, including assessment of conservation strategies. Additional information can be found on our announcement page: https://collections.plos.org/s/biodiversity

If you would like your manuscript to be considered for this collection, please let us know in your cover letter and we will ensure that your paper is treated as if you were responding to this call. If you would prefer to remove your manuscript from collection consideration, please specify this in the cover letter.

Reviewers' comments:

Reviewer's Responses to Questions

**Comments to the Author**

1. Is the manuscript technically sound, and do the data support the conclusions?

Reviewer #1: Partly

Reviewer #2: Partly

2. Has the statistical analysis been performed appropriately and rigorously? 

Reviewer #1: Yes

Reviewer #2: Yes

3. Have the authors made all data underlying the findings in their manuscript fully available?

Reviewer #1: No

Reviewer #2: No

4. Is the manuscript presented in an intelligible fashion and written in standard English?

Reviewer #1: Yes

Reviewer #2: Yes

5. Review Comments to the Author

Reviewer #1: The manuscript is useful in illustrating the dangers of, as the authors call them "snap-shot" surveys without careful consideration of sample size needs and the biology of the species. The authors used camera photos from 20 paired camera traps in 2013-2014 to estimate density of jaguar in Belize. To explore the robustness of their estimates, they subsampled the year data using a rolling consecutive window of 90 and 180 day increments. They found that density estimates were not temporally robust and recommended that when samples sizes are insufficient from a traditional "snap-shot" camera survey of 90 days, researchers use a longer period of sampling to discover any idiosyncrasies in estimates that compromise the robustness of their estimates.

A general recommendation I have is to reword some parts from the beginning to make clearer that the issue at the core of the lack of robustness in the estimates was sample size. As well, to point out that subsampling consecutively throughout the year is not representative of a single density estimate if biological or demographic factors cause changes in density on the study area (e.g., births, deaths, increase in transients, more rigid territorial boundaries pushing individuals to areas outside of the study area during mating season, etc). This is discussed in the Discussion, but deserves space in the Introduction. For example, in lines 74-78 in the Introduction, though you do qualify the statement with "few individuals," a careless reader might assume you are suggesting that 3 month surveys are generally inadequate for SCR. But the reason it is inadequate here, and for other elusive and rare species, is that you often cannot obtain sufficient numbers of spatial recaptures to accurately estimate sigma and consequently density. The second sentence is also misleading. We should not consider a 90 day snap-shot estimate as a single estimate from a distribution of estimates for the year. If density changes throughout the year on the study area (for example, during the mating period in Nov-Jan), then data from April-June is a sample drawn from a different "population" than Nov-Jan.

I also question the conclusion that sigma was overestimated when compared to the GPS derived MCP home range estimates on only 2 individuals. Especially when one male was only tracked for 202 days in the year. It very well could suggest that sigma was positively biased, but the data are inadequate to conclude that definitively, and because of this I feel that it deserves less space in the body and the inadequacies of the sample size for comparison should be mentioned explicitly. Also, since both collared males were detected on the camera traps, the maximum observed recapture distances might be of interest if they had spatial recaptures on camera.

There are a few methods and results that I would recommend be included. For methods, I could recommend including: the period of time between "detections" when individuals were captured in consecutive photos, and the observation model used (e.g., binomial, Poisson). For results, I would recommend including information on the number of spatial recaptures of individuals. SCR estimates require at minimum around 20 spatial recaptures in a session to obtain robust estimates of density.

Overall, aside from the survey map, all figures and tables should be improved for publication for clarity and presentation. I could not read Fig. 3 at all and on Fig. 2, the x-axis text should be clearer (printed diagonally or not every day printed). The table 1 caption should be more descriptive. For example, what is N? I know it is the number of sessions, but this should be clearly defined. And I am guessing "Total" is when the whole period is considered?

My final general comment is that the authors should strive to clearly distinguish between real biological change over the course of the year that may lead to a "lack of robustness" because density is truly changing and a true lack of robustness arising from sampling insufficiencies. In the former case, the "lack of robustness" is expected and should be accounted for in survey design. Now, if you are recommending researchers lengthen their survey period and subsample as was done in this manuscript because it will help them understand true process variation in density, then this should be said more explicitly. In the latter case, if intensive surveying is infeasible, then the conclusions of the authors are practical. Setting out a single camera array for a year does minimize field effort and costs, and might elucidate any strange results arising from sampling insufficiencies, although if that design does not obtain a sufficient sampling size during any of the 90 day periods throughout the year, then the problem would not be resolved.

Line details:

Lines 84-88: The example used non-spatial CR. The authors should mention this, as non spatial CR estimates are even less robust for wide-ranging species than SCR.

Lines 270-271: The linear R2 is reported twice. Should one be curvilinear?

Lines 380-390: You conclude that the variations in density estimates come from variations in ranging behavior. This assumes that your estimates of sigma are robust and accurate. A third option is that the number of spatial recaptures was inadequate to accurately estimate sigma. This may be what you are describing in option 1, but this is unclear as written.

Lines 423-425: If territories become increasingly overlapped or territorial, and/or if transients move onto or off of the study area, then "density" over that area is truly changing, and abundance as well, if you are holding a specific area under consideration constant. This is why "snap-shot" surveys should be chosen to minimize change over the course of the survey for the species at hand and why density estimates are specific to that area in that time period and should be repeated at the same time every year for monitoring.

Lines 446-448: I am glad the authors included this sentence, as it is relevant to many of my other comments, but this should be clearer prior to the discussion.

Lines 454-456: this could be said of any sample survey.

Reviewer #2: Johnson et al. conducted a camera trap survey and individually identified Jaguars in Belize from the pictures. They conducted the study for one entire year over a relatively small area (i.e. 490km2) with relatively few paired cameras (20) placed on trails. They used individual detections and applied SCR models to estimate density of males Jaguars. They especially tested whether the length and timing of the period used to collect data could influence the density obtained from the SCR model. This is an interesting question as the closure assumption is required for CR types models, but might be difficult to respect in reality. Indeed, the amount of data necessary for the models requires long data collection period. The authors have done a good job in maintaining camera traps for full year and present an interesting method to check how SCR density estimates can vary over time. I agree that comparisons of density estimates obtained at different time, in different study areas is difficult to interpret, especially for studies with a small sample size that can strongly be influenced by stochastic events in the population. However, there are a number of points and conclusions in the manuscript that would need to be reworded or clarified, in my opinion.

General comments:

I do not understand why conclude that L.31“current use of one-off (‘snap-shot’) 3-month surveys is inadequate for accurate, precise and robust density estimation”. When I look at the different estimates provided in the figure 3A, I see that both curves from the 90 or the 180 days sessions have relatively similar pattern. As we can expect, the density estimates of the 180-day session is lot less variable (due to more samples and longer period) than the 90-day session, but estimates are similar with CI overlapping between estimates of the two periods.

To my opinion, it is difficult to conclude that one data collection period is better (inadequate) than the other from the results presented here. I would rather suggest that estimates from the 90/180 days period are different snapshot representations of the densities as perceived by the model. Indeed, since we do not know the true abundance it is difficult to say what method is the best.

In the case of studies with small sample size, results are a lot more sensitive to addition of new detections of individuals in the dataset (e.g. caused by immigration, dispersal event of an individual crossing the study area). Because the 180-day session has more detections, the results are less sensitive to small changes in the population and new detection of individuals. However, since the collection period is longer it is a lot more prone to violate the closure assumption. As Dupont et al 2018 (cited in the manuscript) showed, consequences of violating the closure assumption can be severe if the sampling period overlap with a high birth (or immigration) pulse.

Since we do not know the true population size it is difficult to say whether a sudden increase in the estimates as perceived in the figure 3A is true perception of the reality. Maybe it is, and limitations of the study (e.g. assume constant probability of detection, location of cameras only on trail, ect) do not allow to reveal the true density estimates?

Therefore, given the results presented and the fact that we do not know not the true abundance, I do not think it is possible to choose which method is more “adequate for accurate, precise and robust density estimation”(L.31).

I guess one way to better understand the status of a population is to understand the changes in population dynamics rather than simply comparing static estimates of density. Therefore, the use open population models (Open Population SCR) where vital rates are estimated could be a possibility? Maybe Open Population SCR model is something that the authors could mention?

Detailed comments:

L.23 “robust” is robust the right term? Wouldn’t be “stable” be more appropriate? We don’t know if the estimates are really “robust” since we do not know what the true abundance is?

L.27 “Variation in density for both 90-day and 180-day sessions was almost fully attributable to variation in σ.” Is a reduction of HR size the cause of changes in density? or the result of a chance in density?

L.31 “inadequate” this term is strong. See my general comment.

L.32 “larger”, larger than what?

L.39-52. Not a single reference in this first paragraph of introduction. I think it would help to justify some of the text with references.

L.50. Another important parameter is population size. The larger the population size (large sample size in terms of number of individuals), the less the model will be sensitive to stochastic events.

L.68 “unstable” I am not sure I understand the use of unstable here.

L.75 “three months” why three months here?

L.75 “Estimates derived from three months period with few individuals cannot be considered temporally robust, with likely considerable spatial redistributions after the sample periods.” Please develop as I do not understand why it cannot be considered temporally robust?

L.76 I do not understand this sentence: “Such estimates should be considered a single estimate from a wider distribution of possible density estimate outcomes throughout a year”, can you please explain what is “a wider distribution of possible density estimates”? I would find it normal that density estimates fluctuate throughout a year with animals moving in a out of the study area.

L.89. “robust” again, wouldn’t stable be a better term here?

L.114. “social structure” what kind of variables test for this? I only see the number of individuals detected as a variable? the sex-ratio would be a better test for the social structure, no?

L.118 “behavioral idiosyncrasies” I am not sure what kind of behavior we could highlight with this type of test, could you please explain?

L.151. is the day 1, the firth of March? Maybe instead of the x-axis being the density estimation ”run” in fig 2 and 3, it could be the first day of the period? With a similar x-axis between the 90-days and 180days plot, it would be easier to compare the estimates.

L.153. it is not clear to me what was the spatial domain used in the SCR model, is it the area within the black polygon in fig 1? It might be good to show what was considered as buffer and spatial domain in the fig1.

Additionally, how was defined one detection of a jaguar? Can it be multiple detections per day? Did you consider binary detection (since use g0) with each day being an “occasion”? this info is missing in the methods.

L.167 why not running sex-specific models (no covariate needed) and show the results on females as well? I think it would be interesting as the need to lengthing the study period maybe more important for females than males, given the lower sample size?

L.178. I do not understand how a change in g0 could influence density and be the results of demographic factors.

L.217. Figure 2, I do not understand what represent this figure? How does it differ from figure 3a?

L.260. I am not really sure what is the point of showing a relation between g0 and number of detections?

L.334. “The cause of variation…” why wouldn’t such variation be a possible representation of the reality, wouldn’t it be possible that for a period of time, the density doubled in the study area due to stochasticity in space use of some individuals (suddenly many individuals in the study area)? My point is that it is a dynamic system and we could expect local changes in the distribution of the individuals to modify considerably the density estimates.

L.350 “We infer that a positively-biased estimate of σ will equate to a negatively-biased density estimate, and vice versa” This is a very strong assertion. I would suggest toning it down. Can’t sigma be larger with a larger density estimates? This might mean larger HR overlap, but why not?

L.351. There were only 2 GPS collared individuals, maybe it is not representative of the population? Additionally, cameras are only placed at trails with large area in the north west of the area (Fig1) without cameras. This may not adequality capture space usage of jaguars?

L.375. SCR may also better estimate HR size using non-euclidean distance models (Sutherland et al. 2015).

L.404. Why not deciding the collection period based on seasonality? As it seems that male detectability was lower when the wet season progressed?

L.457. Wouldn’t an increase of the “Spatial” sampling would be better? This would allow to overlap with a larger proportion of the population? Therefore, less sensitive to small stochastic events.

L.461. I would recommend the use of simulation in combination with real studies. Because with real studies, we are not able to control parameters and cannot understand with certainty what kind of factors is responsible of a given result.

References:

Sutherland, C., Fuller, A.K. and Royle, J.A. (2015), Modelling non‐Euclidean movement and landscape connectivity in highly structured ecological networks. Methods Ecol Evol, 6: 169-177. doi:10.1111/2041-210X.12316

6. PLOS authors have the option to publish the peer review history of their article (what does this mean?). If published, this will include your full peer review and any attached files.

Reviewer #1: No

Reviewer #2: No

---

## [Author Response · Author response to Decision Letter 0]

16 Mar 2020

Journal Requirements:

1. In your Methods section, please provide additional location information of the study area, including geographic coordinates for the data set if available.

Provided midpoint WSG84 google map reference point for the area in Methods section describing the study area (Lines 169-170)

2. Our internal editors have looked over your manuscript and determined that it is within the scope of our Biodiversity Conservation Call for Papers. This collection of papers is headed by a team of Guest Editors for PLOS ONE (https://collections.plos.org/s/biodiversity). The Collection will encompass a diverse range of research articles on biodiversity conservation, including assessment of conservation strategies. Additional information can be found on our announcement page: https://collections.plos.org/s/biodiversity

If you would like your manuscript to be considered for this collection, please let us know in your cover letter and we will ensure that your paper is treated as if you were responding to this call. If you would prefer to remove your manuscript from collection consideration, please specify this in the cover letter.

We have read the announcement. We prefer not to disrupt the current review process and prefer to continue the current process. It is unknown to us to what extend the collection of papers will increase the impact of our current manuscript. 

We will include the single full year data file available, allowing researchers to create their own separations. 

 

Reviewer #1: 

The manuscript is useful in illustrating the dangers of, as the authors call them "snap-shot" surveys without careful consideration of sample size needs and the biology of the species. The authors used camera photos from 20 paired camera traps in 2013-2014 to estimate density of jaguar in Belize. To explore the robustness of their estimates, they subsampled the year data using a rolling consecutive window of 90 and 180 day increments. They found that density estimates were not temporally robust and recommended that when samples sizes are insufficient from a traditional "snap-shot" camera survey of 90 days, researchers use a longer period of sampling to discover any idiosyncrasies in estimates that compromise the robustness of their estimates.

A general recommendation I have is to reword some parts from the beginning to make clearer that the issue at the core of the lack of robustness in the estimates was sample size. As well, to point out that subsampling consecutively throughout the year is not representative of a single density estimate if biological or demographic factors cause changes in density on the study area (e.g., births, deaths, increase in transients, more rigid territorial boundaries pushing individuals to areas outside of the study area during mating season, etc). This is discussed in the Discussion, but deserves space in the Introduction. 

We have rectified this point by adding sections in the Introduction, and adding the importance of larger samples of recaptures. Specifically lines 87-92, lines 103-114.

For example, in lines 74-78 in the Introduction, though you do qualify the statement with "few individuals," a careless reader might assume you are suggesting that 3 month surveys are generally inadequate for SCR. But the reason it is inadequate here, and for other elusive and rare species, is that you often cannot obtain sufficient numbers of spatial recaptures to accurately estimate sigma and consequently density. The second sentence is also misleading. We should not consider a 90 day snap-shot estimate as a single estimate from a distribution of estimates for the year. If density changes throughout the year on the study area (for example, during the mating period in Nov-Jan), then data from April-June is a sample drawn from a different "population" than Nov-Jan.

We agree with the lack of clarity regarding this, and the misleading way of indicating this. We have now indicated our intentions more clearly and expanded in the Introduction on the purpose of our study. Specifically lines 87-92, lines 95-114

I also question the conclusion that sigma was overestimated when compared to the GPS derived MCP home range estimates on only 2 individuals. Especially when one male was only tracked for 202 days in the year. It very well could suggest that sigma was positively biased, but the data are inadequate to conclude that definitively, and because of this I feel that it deserves less space in the body and the inadequacies of the sample size for comparison should be mentioned explicitly. Also, since both collared males were detected on the camera traps, the maximum observed recapture distances might be of interest if they had spatial recaptures on camera.

We have now included the maximum distances moved by all 21 observed individuals. The two collared individuals ranked 2 and 3 in this list, indicating that they were among the widest ranging individuals in the sample of males, so strengthening our argument that sigma was overestimated when compared to GPS data. We have now stressed the low sample size of collared individuals and indicated that further study is needed. This subject has not been broached in SECR literature and requires more attention as GPS data allows one of the few means of validation of sigma. We hope this study encourages further reporting on this. Specifically lines 254-261, lines 429-435, lines 480-483

There are a few methods and results that I would recommend be included. For methods, I could recommend including: the period of time between "detections" when individuals were captured in consecutive photos, and the observation model used (e.g., binomial, Poisson). For results, I would recommend including information on the number of spatial recaptures of individuals. SCR estimates require at minimum around 20 spatial recaptures in a session to obtain robust estimates of density.

-Methods: We have now indicated the period of time between detections per camera that we used as independent capture records and we have referred to the use of ‘secr’ in R for the observational model. Specifically lines 193-194, line 205. 

-Results: We have now included the number of spatial recaptures per session and the mean number of spatial recaptures per individual per session. All sessions exceed 20 spatial recaptures. Many thanks for the suggestion as we identified some interesting relationships and have now included them in the manuscript. Specifically lines 208-210, line 338, Figure 4, lines 346-347, lines 373-380, lines 422-424.

Overall, aside from the survey map, all figures and tables should be improved for publication for clarity and presentation. I could not read Fig. 3 at all and on Fig. 2, the x-axis text should be clearer (printed diagonally or not every day printed). The table 1 caption should be more descriptive. For example, what is N? I know it is the number of sessions, but this should be clearly defined. And I am guessing "Total" is when the whole period is considered?

-Figure 1: We have improved the map by indicating where the Cockscomb Basin is located in relation to the country map of Belize. 

-Figure 2, 3, and 4: We have increased the gaps between the x-axis labels, which are now spaced by 20 sessions instead of 5. 

-Figure 3: We have split this figure into two figures (now Figure 3 and 4new) and included the variation of male spatial recaptures within one of the figures (Figure 4new), so that each figure now has 3 panels only. 

-Figure 4original: is now split into two figures (now Figure 5 and 6). We have removed four of the panels and split the remaining four panels into the two new figures. Figure 5 is a scatterplot of density and sigma, while Figure 6 is a scatterplot of density and number of female individuals and detections, including separate colouring for the first 160 90-day sessions, as per previous.

-Figure 7: We added a photo of a jaguar mating event from the survey grid during the survey period. 

-Table 1: We have changed the labels indicating that Total is the full dataset of locations and indicated N as number of sessions. We have removed the row of sigma as this is clear in Figure 3.

My final general comment is that the authors should strive to clearly distinguish between real biological change over the course of the year that may lead to a "lack of robustness" because density is truly changing and a true lack of robustness arising from sampling insufficiencies. In the former case, the "lack of robustness" is expected and should be accounted for in survey design. Now, if you are recommending researchers lengthen their survey period and subsample as was done in this manuscript because it will help them understand true process variation in density, then this should be said more explicitly. In the latter case, if intensive surveying is infeasible, then the conclusions of the authors are practical. Setting out a single camera array for a year does minimize field effort and costs, and might elucidate any strange results arising from sampling insufficiencies, although if that design does not obtain a sufficient sampling size during any of the 90 day periods throughout the year, then the problem would not be resolved.

Many thanks for the indication that this is not clear. We have rewritten the Introduction and Discussion, making more explicit the need for lengthening surveys to understand process variation as the reviewers suggests. 

Line details:

Lines 84-88: The example used non-spatial CR. The authors should mention this, as non spatial CR estimates are even less robust for wide-ranging species than SCR.

We have added a reference from an MSc thesis from one of our students who analysed the same dataset, using SCR. Specifically line 119.

Lines 270-271: The linear R2 is reported twice. Should one be curvilinear?

Thanks for noting this, we have changed the R2 regression values to correlations and corrected the missing information regarding individuals and detections. Specifically lines 358-360.

Lines 380-390: You conclude that the variations in density estimates come from variations in ranging behavior. This assumes that your estimates of sigma are robust and accurate. A third option is that the number of spatial recaptures was inadequate to accurately estimate sigma. This may be what you are describing in option 1, but this is unclear as written.

We have now included the number of spatial recaptures, which are above the minimum recommended in the literature. We discuss this further in the Discussion. Specifically lines 527-528, lines 533-535.

Lines 423-425: If territories become increasingly overlapped or territorial, and/or if transients move onto or off of the study area, then "density" over that area is truly changing, and abundance as well, if you are holding a specific area under consideration constant. This is why "snap-shot" surveys should be chosen to minimize change over the course of the survey for the species at hand and why density estimates are specific to that area in that time period and should be repeated at the same time every year for monitoring.

We do not agree that this applies for species that are low density and wide-ranging. Repeating at the same time each year for species like jaguars does not sufficiently minimise change between the years (see cited references [9, 10] in the manuscript) as there is too much stochasticity in local conditions (biotic and abiotic) and individual responses; and sample sizes that are too small to balance these. Specifically lines 103-114, line 119.

Lines 446-448: I am glad the authors included this sentence, as it is relevant to many of my other comments, but this should be clearer prior to the discussion.

We have expanded the Introduction to make this point clearer.

Lines 454-456: this could be said of any sample survey.

We agree but this is rarely recognised in the literature and never addressed for elusive wide-ranging carnivores. thus we stress it here.

 

Reviewer #2: 

Johnson Harmsen et al. conducted a camera trap survey and individually identified Jaguars in Belize from the pictures. They conducted the study for one entire year over a relatively small area (i.e. 490km2) with relatively few paired cameras (20) placed on trails. They used individual detections and applied SCR models to estimate density of males Jaguars. They especially tested whether the length and timing of the period used to collect data could influence the density obtained from the SCR model. This is an interesting question as the closure assumption is required for CR types models, but might be difficult to respect in reality. Indeed, the amount of data necessary for the models requires long data collection period. The authors have done a good job in maintaining camera traps for full year and present an interesting method to check how SCR density estimates can vary over time. I agree that comparisons of density estimates obtained at different time, in different study areas is difficult to interpret, especially for studies with a small sample size that can strongly be influenced by stochastic events in the population. However, there are a number of points and conclusions in the manuscript that would need to be reworded or clarified, in my opinion.

General comments:

I do not understand why conclude that L.31“current use of one-off (‘snap-shot’) 3-month surveys is inadequate for accurate, precise and robust density estimation”. When I look at the different estimates provided in the figure 3A, I see that both curves from the 90 or the 180 days sessions have relatively similar pattern. As we can expect, the density estimates of the 180-day session is lot less variable (due to more samples and longer period) than the 90-day session, but estimates are similar with CI overlapping between estimates of the two periods.

To my opinion, it is difficult to conclude that one data collection period is better (inadequate) than the other from the results presented here. I would rather suggest that estimates from the 90/180 days period are different snapshot representations of the densities as perceived by the model. Indeed, since we do not know the true abundance it is difficult to say what method is the best.

In the case of studies with small sample size, results are a lot more sensitive to addition of new detections of individuals in the dataset (e.g. caused by immigration, dispersal event of an individual crossing the study area). Because the 180-day session has more detections, the results are less sensitive to small changes in the population and new detection of individuals. However, since the collection period is longer it is a lot more prone to violate the closure assumption. As Dupont et al 2018 (cited in the manuscript) showed, consequences of violating the closure assumption can be severe if the sampling period overlap with a high birth (or immigration) pulse.

Since we do not know the true population size it is difficult to say whether a sudden increase in the estimates as perceived in the figure 3A is true perception of the reality. Maybe it is, and limitations of the study (e.g. assume constant probability of detection, location of cameras only on trail, ect) do not allow to reveal the true density estimates?

Therefore, given the results presented and the fact that we do not know not the true abundance, I do not think it is possible to choose which method is more “adequate for accurate, precise and robust density estimation”(L.31).

We agree with the statement and have removed the notation of 3 months. As 6 months are not usually used with surveys, and 3 months surveys are almost the norm, the sentence referred to the normative 3-month surveys. We have removed the reference to 3 months and changed to short-term to indicate the inadequacy of one-off surveys as representative for an area. Specifically line 33.

I guess one way to better understand the status of a population is to understand the changes in population dynamics rather than simply comparing static estimates of density. Therefore, the use open population models (Open Population SCR) where vital rates are estimated could be a possibility? Maybe Open Population SCR model is something that the authors could mention?

We have added to the Discussion that open population SCR models would provide useful quantifications for modelling and cite a paper that uses data from the current long-term jaguar dataset used in this study. Specifically lines 497-500, lines 619-621.

Detailed comments:

L.23 “robust” is robust the right term? Wouldn’t be “stable” be more appropriate? We don’t know if the estimates are really “robust” since we do not know what the true abundance is?

We feel that robust is the right term. Any change in density estimates should reflect true population change. If density estimates depend on spatial and temporal dimensions of measurement then it cannot be considered robust in space and time.

L.27 “Variation in density for both 90-day and 180-day sessions was almost fully attributable to variation in σ.” Is a reduction of HR size the cause of changes in density? or the result of a chance in density?

We have edited the Abstract to make this clearer. Specifically lines 26-29, lines 33-40.

L.31 “inadequate” this term is strong. See my general comment.

We have removed the term ‘inadequate’, see response within general comment, specifically line 34.

L.32 “larger”, larger than what?

Thank you for noting this, we have now compared this to what we detected in this study, which is similar or larger than published estimates from camera trap studies of large wide-ranging carnivores. Specifically lines 37-40.

L.39-52. Not a single reference in this first paragraph of introduction. I think it would help to justify some of the text with references.

We have now added references. Specifically line 49, line 54, line 58.

L.50. Another important parameter is population size. The larger the population size (large sample size in terms of number of individuals), the less the model will be sensitive to stochastic events.

Many thanks for noting, we have changed this. Specifically lines 57-58.

L.68 “unstable” I am not sure I understand the use of unstable here.

We removed the word “unstable” here and expanded the narrative. Specifically 81-92.

L.75 “three months” why three months here?

We changed this to “2-3 month period, as per Karanth et al. to emphasise the link with the previous narrative, indicating that 2-3 months has become accepted as the standard period for camera trapping of large carnivores to ensure population closure. Specifically lines 79-80, lines 

L.75 “Estimates derived from three months period with few individuals cannot be considered temporally robust, with likely considerable spatial redistributions after the sample periods.” Please develop as I do not understand why it cannot be considered temporally robust?

We have now clarified this in the narrative. Specifically lines lines 64-67, lines 83-92, lines 103-114. 

L.76 I do not understand this sentence: “Such estimates should be considered a single estimate from a wider distribution of possible density estimate outcomes throughout a year”, can you please explain what is “a wider distribution of possible density estimates”? I would find it normal that density estimates fluctuate throughout a year with animals moving in a out of the study area.

We have expanded this section and explained. Specifically lines 100-114.

L.89. “robust” again, wouldn’t stable be a better term here?

See previous

L.114. “social structure” what kind of variables test for this? I only see the number of individuals detected as a variable? the sex-ratio would be a better test for the social structure, no?

We assess social structure as the number of males and females, and the number of detections of males and females. Specifically line 158

L.118 “behavioral idiosyncrasies” I am not sure what kind of behavior we could highlight with this type of test, could you please explain?

We have deleted ‘behavioural’ and explained that we test for social changes by assessing the number and detection rate of males and females. Specifically lines 157-158. 

L.151. is the day 1, the firth of March? Maybe instead of the x-axis being the density estimation ”run” in fig 2 and 3, it could be the first day of the period? With a similar x-axis between the 90-days and 180days plot, it would be easier to compare the estimates.

Each run (now called session) refers to a period of 90 or 180 days. The use of dates on the x-axis clutters and confuses the figure. We have improved the clarity of this figure taking by following advice of Reviewer 1 ; and we have added to the legend that for the 90 days, session 1 represents 19-Mar to 16-Jun-2013 and session 276 represents 19-Dec-2013 to 18-Mar-2014, etc. Specifically lines 304-307, lines 343-349.

L.153. it is not clear to me what was the spatial domain used in the SCR model, is it the area within the black polygon in fig 1? It might be good to show what was considered as buffer and spatial domain in the fig1.

As indicated in the legend on the figure, the black polygon delineates the boundary of the protected area. We have not included the buffer in Figure 1 as areas this would need to be viewed at a different spatial scale, reducing detail of the camera and trail area. We include in the Methods the we used a buffer of 30 km to define the area of interest (mask). Specifically lines 205-207.

Additionally, how was defined one detection of a jaguar? Can it be multiple detections per day? Did you consider binary detection (since use g0) with each day being an “occasion”? this info is missing in the methods.

We excluded repeat detection of the same jaguar on the same day at the same camera location. We have now included this information in the Methods. Specifically lines 193-194.

L.167 why not running sex-specific models (no covariate needed) and show the results on females as well? I think it would be interesting as the need to lengthing the study period maybe more important for females than males, given the lower sample size?

We agree that this would be interesting, however the sample size of females (number of individuals detected, and number of spatial recaptures) is not sufficiently large to estimate D, g0 and σ using SCR

L.178. I do not understand how a change in g0 could influence density and be the results of demographic factors.

Density will be influenced by the number of individuals detected in the survey grid and/or how far they move within the survey grid. If the number of individuals stays the same through time but the extent of their ranges change, then sigma will vary through time with density. If the number of individuals changes through time, but the extent of their ranges remains the same, then the number of individuals and g0 will change with density. We refer to changes in number of individuals and capture probability as demographic/social factors. Specifically lines 226-232.

L.217. Figure 2, I do not understand what represent this figure? How does it differ from figure 3a?

In Fig 2, the mean density estimates are sorted from low to high; in Fig 3 (and the new figure Fig 4) the density estimates are displayed as a time series (consecutive estimates through time). We use Fig 2 to illustrate the range in density estimates obtained over the year; this is clearer when displayed from lowest to highest rather than as a time series. Early in-house reviewing indicated that colleagues underestimated the variation Fig 3 (when density estimates were displayed as a time series). 

L.260. I am not really sure what is the point of showing a relation between g0 and number of detections?

As indicated for the previous comment of line 178, we are looking for factors which covary with the seasonal fluctuation of density across the year. Specifically lines 226-232.

L.334. “The cause of variation…” why wouldn’t such variation be a possible representation of the reality, wouldn’t it be possible that for a period of time, the density doubled in the study area due to stochasticity in space use of some individuals (suddenly many individuals in the study area)? My point is that it is a dynamic system and we could expect local changes in the distribution of the individuals to modify considerably the density estimates.

Yes, this is exactly what we demonstrate with this manuscript. Density estimates vary over the course of a year within the survey grid due to stochasticity in the interactive space use of the sampled individuals. We have indicated that density estimates have been used in the literature for meta-analyses assuming static robust estimates. Specifically lines 596-601.

L.350 “We infer that a positively-biased estimate of σ will equate to a negatively-biased density estimate, and vice versa” This is a very strong assertion. I would suggest toning it down. 

We have changed the text to ‘.. a low estimate of sigma will equate to a high estimate of density..’. Specifically lines 476-478.

Can’t sigma be larger with a larger density estimates? This might mean larger HR overlap, but why not?

In principle, this is possible; however in this study our data showed a strong inverse relationship between density and sigma. Specifically lines 367-372.

L.351. There were only 2 GPS collared individuals, maybe it is not representative of the population? Additionally, cameras are only placed at trails with large area in the north west of the area (Fig1) without cameras. This may not adequality capture space usage of jaguars?

We have indicated the low sample size of the two collared individuals. As a comparison with the spatial recaptures of the detected individuals, we have indicated the maximum distances moved between camera stations for all 21 observed individuals. The two collared individuals ranked 2 and 3 in this list, indicating that they were among the widest ranging individuals sampled, strengthening our argument. Specifically lines 254-261, lines 429-435, lines 480-483.

L.375. SCR may also better estimate HR size using non-euclidean distance models (Sutherland et al. 2015).

Thank you for this reference. We have now included the use of non-euclidean distance models as another alternative to the traditional set of detection functions. Specifically lines 516-518.

L.404. Why not deciding the collection period based on seasonality? As it seems that male detectability was lower when the wet season progressed?

We highlight a potential seasonal or climatic trend, but there is insufficient evidence to conclude that the observed temporal variation are entirely seasonal. This would require several years of data collection over multiple seasons. Specifically lines 497-504, line 575, line 610.

L.457. Wouldn’t an increase of the “Spatial” sampling would be better? This would allow to overlap with a larger proportion of the population? Therefore, less sensitive to small stochastic events.

This is indicated in the Discussion in the lines ……” By sampling more individuals, with larger survey areas and/or denser camera grids, the sampling sessions may become more robust to temporal variation in carnivore behaviour. However, logistical constraints have long limited camera trap surveys, with most camera studies of jaguars and other large carnivores failing to sample more than 10 individuals [1,4]”. Specifically lines 608-613.

L.461. I would recommend the use of simulation in combination with real studies. Because with real studies, we are not able to control parameters and cannot understand with certainty what kind of factors is responsible of a given result.

We agree that individual-based models may offer the level of complexity required to simulate stochastic behaviour of some species. However, we still know little about the behavioural ecology of elusive large carnivores such as jaguars, therefore we believe at this stage, empirical field studies are required. However, we have deleted ‘.rather than the use of simulation studies..’. Specifically lines 621-622. 

References:

Sutherland, C., Fuller, A.K. and Royle, J.A. (2015), Modelling non‐Euclidean movement and landscape connectivity in highly structured ecological networks. Methods Ecol Evol, 6: 169-177. doi:10.1111/2041-210X.12316

 

6. PLOS authors have the option to publish the peer review history of their article (what does this mean?). If published, this will include your full peer review and any attached files.

The authors are fine with the review process and comments being publicly available for readers.

---

## [Decision Letter · Decision Letter 1]

14 Apr 2020

PONE-D-19-35092R1

Spatially explicit capture recapture density estimates: robustness, accuracy and precision in a long-term study of jaguars (Panthera onca)

PLOS ONE

Dear Dr. Harmsen,

Thank you for submitting your manuscript to PLOS ONE. After careful consideration, we feel that it has merit but does not fully meet PLOS ONE’s publication criteria as it currently stands. Therefore, we invite you to submit a revised version of the manuscript that addresses the points raised during the review process.

Both previous reviewers have now reviewed your revised manuscript. They both noted an improvement in the quality of the ms and thank the authors for answering and/or taking into account their comments. 

They still have some minor points requiring your comments (some clarifications and suggestions about sentences). Please take it in consideration, and in particular, be careful about your words in the conclusion suggesting that spatial capture-recapture may not be a useful method in general, a statement which seems quite strong regarding your study.

We would appreciate receiving your revised manuscript by May 29 2020 11:59PM. To enhance the reproducibility of your results, we recommend that if applicable you deposit your laboratory protocols in protocols.io, where a protocol can be assigned its own identifier (DOI) such that it can be cited independently in the future. For instructions see: http://journals.plos.org/plosone/s/submission-guidelines#loc-laboratory-protocols

We look forward to receiving your revised manuscript.

Kind regards,

Guillaume Souchay

Academic Editor

PLOS ONE

Reviewers' comments:

Reviewer's Responses to Questions

**Comments to the Author**

1. If the authors have adequately addressed your comments raised in a previous round of review and you feel that this manuscript is now acceptable for publication, you may indicate that here to bypass the “Comments to the Author” section, enter your conflict of interest statement in the “Confidential to Editor” section, and submit your "Accept" recommendation.

Reviewer #1: (No Response)

Reviewer #2: (No Response)

2. Is the manuscript technically sound, and do the data support the conclusions?

Reviewer #1: Partly

Reviewer #2: Partly

3. Has the statistical analysis been performed appropriately and rigorously? 

Reviewer #1: Yes

Reviewer #2: Yes

4. Have the authors made all data underlying the findings in their manuscript fully available?

Reviewer #1: No

Reviewer #2: No

5. Is the manuscript presented in an intelligible fashion and written in standard English?

Reviewer #1: Yes

Reviewer #2: Yes

6. Review Comments to the Author

Reviewer #1: The manuscript is much improved, thank you for addressing the previous comments. Following are some new comments and suggestions that have come to mind with the revisions:

1. The number of spatial recaptures seems high for such an elusive species, and especially considering the influence that the change of one occasion had on the 90 day sampling periods. I just want to make sure it is number of spatial recaptures, not the number of recaptures? If individual i is detected a total of 5 times, but 2 of those detections are at the same camera, consecutively, then you have only obtained 3 spatial recaptures, for example. And the traps were unbaited?

2. Thinking again about the variation in density; is a difference of 2 jaguars / 100 km2 very large? Perhaps so, given the species in question. But your sampling area is 120 km2. If individuals are moving on and off the sampling grid throughout the year or use trails more or less because of seasonal variation in space use, it seems feasible that the estimated density of jaguars/100km2 could actually vary by that amount.

3. Line 106: "local" population is unclear. Did you mean the population in the desired area of inference or the sampled area? It makes more sense that the variation was due to variation in estimated density across the sampling grid because of the small sampling area, even though the population in the larger study area was not changing.

4. Lines 119-122 (the semi-colon is unnecessary here), it is unclear what is meant by sigma and g0 are directly related to the abundance and activity of animals in traditional CR. Both are related to the detection probability of an individual in CR, as larger scales of movements translate to lower detectability, just as lower g0 would. What is meant by "activity" in traditional CR, and how are the parameters directly related to abundance?

5. I did not mention this in my last review, but a potential for sampling bias because of only sampling on trails should probably be mentioned in the methods or discussion. Could there be seasonal variation in trail use? Especially during mating season, as females use trails less?

6. Fig 1 is improved, but for Fig 2-4, there is still a problem of resolution. The numbers and lines look fuzzy. I also cannot tell the difference between dashed and solid lines. For Fig. 4, I don't understand the reasons for putting two lines in one graph with 2 different y axes. This makes it difficult to read. Your descriptions for b and c for Fig. 4 also do not match the figure.

7. I would recommend putting the sigma row back in Table 1. It is much easier for the reader to quickly see in the Table the differences between the sigma and collar estimates. And, in the body you could refrain from repeating the exact estimates, just reference the Table 1, and describe the differences.

8. I think it's important to mention in the discussion or methods that the validity of using sigma as a measure of home range size relies on the detection process mirroring the half normal detection function. I don't really trust the direct comparison of home range estimators using the formula derived from the half normal function in general, and especially because of placement of cameras on trails. Essentially, the detection process is sampling the way jaguars move on trails but the collars are sampling movement in general. The MDM of male 1 is large, for example, compared to what would be the diameter of a circular home range based on the 100% MCP. Also, in addition to qualifying the use of this comparison (my reasonings stated were taken from reference 14 of your paper, so no further references needed), you would actually need to use 95% home ranges with that formula, if you are assuming the probability distribution of MCP points mirrors the distribution of the half normal.

9. Overall, be consistent with tenses. And I saw at least once where sigma was spelled out rather than the symbol used (line 229).

10. line 364: "decreased with" is repeated.

11. lines 416-17 Again, I would just be careful with wording and interpretation of density change throughout the year. It is a strong statement to say "we cannot consider the method robust for low-density, wide-ranging species when using one-off, short-term, small-scale camera trap surveys." Firstly, I would not expect most sampling designs for SCR to be robust across the year, but most sampling designs are also not meant to be, as they balance logistical feasibility with the assumption of *closed* periods based on known biology and seasonal space use of the species. SCR density estimates would only be identical from Jan-December on a static trapping array if the species was demographically closed to deaths and births and did not have seasonal variation in space use, or it was demographically closed and sampling accommodated any strong seasonal violations of geographic closure or heterogeneous use of space across the year. And, of course, on the assumption that sampling sizes are sufficient to capture a range of animal movements.

For example, elk have sharply contracted group ranges during the rutting season, and managers sometimes specifically sample for SCR during this time period to capture the entire herd within a logistically feasible sampling extent, but density (which is higher than normal) is also interpreted in this context and if the same trapping array was used for the whole year, density would indeed fluctuate greatly. This is just an exaggerated example of why we would not expect SCR estimates from a static trapping array to be "robust" across the year, even if it is quite accurately estimating true densities.

A corollary to jaguar might be that at certain times of the year, territories overlap and shrink more than other times. Or jaguars use trails more in a certain season. But if the researcher is aware of this, designs a survey and interprets results accordingly, and has a sufficient sample size, then I would consider the study valid and accurate. 3-month oscillations ranging from +/-1 animal from the mean of 2 per 100 km2 seem biologically reasonable to me, but I am not a jaguar expert, so perhaps you could discuss if it really is indicative of SCR not accurately estimating density throughout the year. Especially because reference 10 found a similar fluctuation in density across years, which you refer to in this paper as "widely fluctuating"

I think the more important indicator of a problem is significant change with small variation in start date of the survey, which you did find and covered in the Results-Time series section.

I would lastly like to emphasize that I agree with the benefits of extending the sampling period for elusive, wide-ranging species when more extensive and intensive snap-shot sampling is not logistically feasible. At the very least, it would be useful in a pilot study because it allows the researcher to detect any seasonal differences in space use and adjust monitoring periods accordingly if year round surveys are infeasible every year. I just would be cautious with some of the remaining more strongly worded conclusions because they seem to imply that many prior jaguar (or other SCR) studies are of little value. Or that SCR is inherently flawed, when it would have more to do with poor sampling designs, misuse of the model, and/or misinterpretation of scope of inference.

Reviewer #2: Thank you for addressing my comments and concerns.

I am still not very convinced by the wording of this conclusion:

L.31-34 ”We conclude that one-off (‘snap-shot’) short-term, small-scale camera trap surveys do not sufficiently sample wide-ranging large carnivores for accurate, precise and robust density estimation via SCR».

When reading this sentence, it really seems that we cannot trust results from SCR. However, a small scale “snapshot” SCR study may provide robust and precise estimates of density during the (short) period under study. The observed changes in density estimates may be true reflection of density, making SCR robust in capturing temporal variation in density. But we don’t know this as we do not know the true density estimates. The problem is that the results from small scale studies are difficult to compare over time and across sites as density estimates maybe sensitive to the stochasticity in the behavior of a few individuals.

I would therefore suggest the following (or something similar) instead of a conclusion that looks like SCR are not a good method (as it kind of reads in the conclusion from line 31-34)

Suggestion:

“We conclude that density estimates obtained from one-off (‘snap-shot’) short-term, small-scale camera trap surveys should be carefully interpreted and extrapolated, because different factors, such as temporal stochasticity in behavior of a few individuals, may have strong repercussion on density estimates.“

If changes are made here, i would also recomend changes parts of text where this conclusion is also stated (e.g. L416)

Concerning your response to one of my previous comment:

“Density will be influenced by the number of individuals detected in the survey grid and/or how far they move within the survey grid. If the number of individuals stays the

same through time but the extent of their ranges change, then the number of individuals and g0 will change with density. We refer to changes in number of individuals and capture probability as

demographic/social factors.”.

I still have a hard time with this hypothesis. If their range changes, for example increase, then sigma will increase, but by compensation with the half-normal (Efford and Mowat 2014), g0 will likely decrease. g0 could also change (with or without an increase in sigma) and for all kind of reasons (e.g. Abiotic, Disturbances,...) that are not necessarily directly related to demographic/social factors. I just don’t think it is possible to conclude much about demographic/social factors with a change in g0 in relation with the number of individuals and detections. If this hypothesis is to be stated, more convincing arguments should be given.

Other comments:

L. 29. Please add that it was compared to 2 GPS collared individuals. “…to GPS collar data from two individuals was..».

L.85. “additionally” doesn’t fit here.

L.140. “…by comparing them with….”

L.140. transformed, meaning circular. specify as it looks strange to only see transformation?

L.204. “If the number of individuals stays the same but the extent of their ranges changes, then we would expect density to decrease with increasing spatial recaptures and σ.”

I don’t understand this, if the number of individuals (present in the study area) stays the same, then the density should stay the same, even if number of spatial recaptures increases. Indeed, sigma and p0 have a compensatory pattern (see Efford and Mowat (2014)).

L.206 If the local abundance and detection rates change through time, but the extent of individuals’ ranges remain the same, then we would expect density to increase with increasing number of individuals, detections and g0

I don’t understand this sentence. Local abundance and detection rates should change in which direction of expect density to increase with number of individuals (which individuals? detected individuals?). if detection rate changes, then g0 and detections should change as they should be correlated... I don’t understand the goal of looking at this.

Mechanisms should be explained clearly in order to state this kind of hypothesis.

L.210. Please provide examples of the social mechanisms you would expect to find with the different correlations tested.

L.423. “equate” is still a very strong assertion. I guess it is probably more the high density that could lead to lower sigma estimates.

References

Efford, M., & Mowat, G. (2014). Compensatory heterogeneity in spatially explicit capture—recapture data. Ecology, 95(5), 1341-1348.

7. PLOS authors have the option to publish the peer review history of their article (what does this mean?). If published, this will include your full peer review and any attached files.

Reviewer #1: No

Reviewer #2: No

---

## [Editor Report · Decision Letter 2]

28 May 2020

Spatially explicit capture recapture density estimates: robustness, accuracy and precision in a long-term study of jaguars (Panthera onca)

PONE-D-19-35092R2

Dear Dr. Harmsen,

We are pleased to inform you that your manuscript has been judged scientifically suitable for publication and will be formally accepted for publication once it complies with all outstanding technical requirements.

With kind regards,

Guillaume Souchay

Academic Editor

PLOS ONE
---

## [Editor Report · Acceptance letter]

29 May 2020

PONE-D-19-35092R2 

Spatially explicit capture recapture density estimates: robustness, accuracy and precision in a long-term study of jaguars (Panthera onca) 

Dear Dr. Harmsen:

I am pleased to inform you that your manuscript has been deemed suitable for publication in PLOS ONE. Congratulations! Your manuscript is now with our production department. 

With kind regards,

on behalf of

Dr. Guillaume Souchay 

Academic Editor

PLOS ONE